# MIXED-CURVATURE TREE-SLICED WASSERSTEIN DISTANCE

**Duy-Tung Pham**[*]
FPT Software AI Center
Vietnam
tungpd10@fpt.com

**Viet-Hoang Tran**[*]
Department of Mathematics
National University of Singapore
hoang.tranviet@u.nus.edu

**Thieu N. Vo**
Department of Mathematics
National University of Singapore
thieuvo@nus.edu.sg

**Tan M. Nguyen**
Department of Mathematics
National University of Singapore
tanmn@nus.edu.sg

## ABSTRACT

Mixed-curvature spaces have emerged as a powerful alternative to their Euclidean counterpart, enabling data representations better aligned with the intrinsic structure of complex datasets. However, comparing probability distributions in such spaces remains underexplored: existing measures such as KL divergence and Wasserstein either impose strong distributional assumptions or suffer from high computational costs. The Sliced-Wasserstein (SW) framework offers a tractable alternative for defining distributional distances, but its reliance on one-dimensional projections limits its ability to capture the geometric structure of the ambient space and the input distributions. The Tree-Sliced Wasserstein (TSW) framework provides a principled solution to this limitation by employing tree structures as a richer projected space. Motivated by the intuition that such a space is particularly suitable for representing the geometric properties of mixed-curvature manifolds, we introduce the Mixed-Curvature Tree-Sliced Wasserstein (MCTSW), a novel discrepancy measure that is computationally efficient while faithfully capturing both the topological and geometric structures of mixed-curvature spaces. Specifically, we introduce an adaptation of tree systems and Radon transform to mixed-curvature spaces, which yields a closed-form solution for the optimal transport problem on the tree system. We further provide theoretical analysis on the properties of the Radon transform and the MCTSW distance. Experimental results demonstrate that MCTSW improves distributional comparisons over product-space-based distance and line-based baselines, and that mixed-curvature representations have better performance over constant-curvature alternatives, highlighting their importance for modeling complex datasets.

## 1 INTRODUCTION

Euclidean geometry has long served as the default framework for representing data in machine learning. However, the manifold hypothesis suggests that real-world data typically concentrates on lower-dimensional structures rather than uniformly occupying Euclidean space (Loaiza-Ganem et al., 2024; Gray, 2025). This observation motivates the exploration of alternative geometries, particularly spherical and hyperbolic spaces, which capture structural properties that Euclidean geometry cannot (Cho et al., 2023; Davidson et al., 2018; Dai et al., 2021; Yuan et al., 2025; Park & Kwon, 2019; Chen et al., 2022a). Spherical spaces naturally accommodate directional or periodic data, such as text embeddings (Meng et al., 2019; Banerjee et al., 2022; Xu et al., 2023) or panoramic imagery (Yoon et al., 2022; Sun et al., 2025; Park et al., 2025). In contrast, hyperbolic spaces are especially effective for representing hierarchical structures (Lin et al., 2024b; Chami et al., 2020) and graph-structured data (Zheng et al., 2025; Liu et al., 2019).

---

[*] Co-first authors. Please correspond to: hoang.tranviet@u.nus.edu and tanmn@nus.edu.sg

**Mixed-Curvature Space.** Recent research has begun combining these geometries under the framework of Mixed-Curvature Spaces (MCS), defined as Cartesian products of Euclidean, spherical, and hyperbolic components. MCS provide enhanced flexibility for modeling complex datasets (Skopek et al., 2020; Gu et al., 2019). These hybrid spaces have already demonstrated promise across a broad spectrum of models, ranging from decision trees and random forests (Chlenski et al., 2025), variational autoencoders (Skopek et al., 2020), and graph neural networks (Sun et al., 2022; Wang et al., 2021), to applications in continual learning (Lin et al., 2024a), combinatorial optimization (Liu et al., 2025), and physical sciences (Woodward et al., 2024).

**Optimal Transport.** Despite recent advances, the fundamental challenge of comparing probability distributions across heterogeneous spaces remains unresolved. Standard divergences such as KL divergence are inadequate in this setting, as they are not true metrics and fail to handle distributions with disjoint supports. Optimal Transport (OT) (Villani, 2008; Peyré et al., 2019) provides a geometrically natural metric for comparing probability distributions and has received significant attention in recent years. However, OT suffers from a supercubic complexity in the number of supports, which severely limits its scalability (Peyré et al., 2019). To address this challenge, several scalable variants have been developed, including entropic regularization (Cuturi, 2013), minibatch OT (Fatras et al., 2020), low-rank methods (Forrow et al., 2019; Altschuler et al., 2019; Scetbon et al., 2021), the Sliced Wasserstein distance (Rabin et al., 2012; Bonneel et al., 2015), Tree-Sliced Wasserstein distance (Tran et al., 2025a; 2024c), and Sobolev transport (Le et al., 2022; 2024). Within the Euclidean setting, related methods have also been successfully applied in diverse domains, including machine learning (Kolouri et al., 2018b; Li et al., 2024; Carpintero Perez et al., 2024; Chapel et al., 2025; Lee et al., 2019; Shahbazi et al., 2025; Doan et al., 2021; Wang et al., 2024; Rowland et al., 2019; Melnyk et al., 2024), statistics (Hu & Lin, 2025; Yi & Liu, 2023; Tran et al., 2021; Bonet et al., 2022b; Meunier et al., 2022; Chen & Müller, 2024), computer vision (Arjovsky et al., 2017; Genest et al., 2024; Fujiwara et al., 2024; Li et al., 2022; Yin & Chua, 2024; Bai et al., 2023), and natural language processing (Adhya & Sanyal, 2025; Otao & Yamada, 2023; Luong et al., 2025; Naderializadeh et al., 2021; Dao et al., 2024; Kusner et al., 2015). More general formulations such as Partial Transport (PT) and Unbalanced Optimal Transport (UOT) have likewise found impactful applications in these areas (Séjourné et al., 2022; Kondratyev et al., 2016; Eyring et al., 2024; Gazdieva et al., 2024; Chizat & Bach, 2018; Demetci et al., 2022; Baltrušaitis et al., 2018; Tran et al., 2025d).

**(Tree) Sliced Wasserstein.** Among OT methods, the Sliced Wasserstein (SW) distance has gained prominence by reducing complexity: it projects high-dimensional probability measures onto one-dimensional subspaces, where OT admits closed-form solutions (Rabin et al., 2012; Bonneel et al., 2015). This projection-based strategy has inspired a variety of extensions, including accelerated sampling (Nadjahi et al., 2021; Nguyen et al., 2024a; 2021), informed slicing (Deshpande et al., 2019; Nguyen et al., 2024c; Tran et al., 2024b; Nguyen & Ho, 2023), and generalized integration domains (Kolouri et al., 2019; Quellmalz et al., 2023; Chen et al., 2022b; Bonet et al., 2023b; 2025; 2023c). Yet, the inherent limitation to one-dimensional slices often prevents SW from capturing the full geometric and topological complexity of distributions in curved spaces. The Tree-Sliced Wasserstein (TSW) framework addresses this limitation by replacing linear projections with tree metric systems (Le et al., 2019; Le & Nguyen, 2021; Tran et al., 2025a; Indyk & Thaper, 2003; Takezawa et al., 2021; Lin et al., 2025; Tran et al., 2026b; 2025b; 2026a; 2024d). Unlike lines, tree structures provide richer geometry while still permitting closed-form OT solutions, enabling improved modeling of spatial and topological features. Despite these advances, the potential of TSW in mixed-curvature latent spaces remains largely unexplored, even though such settings require combining heterogeneous geometric properties.

**Contributions.** In this work, we introduce a principled framework for comparing distributions in mixed-curvature spaces. Our main contributions are as follows:

1. In Section 3, we introduce *Mixed-Curvature Trees*, a metric space that admits a tree metric and enables closed-form Wasserstein distances across heterogeneous geometries. We further develop a Radon-type transform on Mixed-Curvature Trees to facilitate efficient distributional comparison.

2. In Section 4, we propose the *Mixed-Curvature Tree-Sliced Wasserstein (MCTSW)* distance. This formulation allows mass to be transported through a tree spanning multiple subspaces, thereby jointly respecting diverse geometric structures.

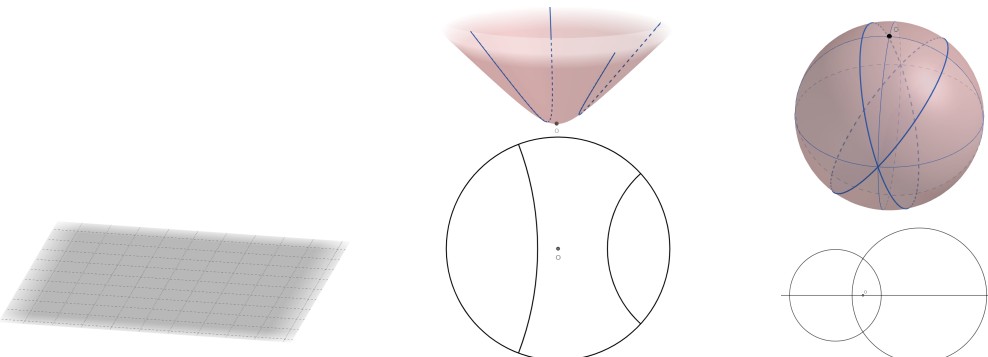

Figure 1: Spaces of constant curvature, the building blocks of mixed-curvature spaces. (Left) Euclidean space $\mathbb{E}_2$, (middle) Hyperbolic spaces $\mathbb{H}_2$ (top) and $\mathbb{P}_2$ (bottom) with corresponding geodesics, and (right) Spherical spaces $\mathbb{S}_2$ (top) and $\mathbb{D}_2$ (bottom) with corresponding geodesics.

3. In Section 5, we validate our framework through experiments on gradient flow, variational autoencoders, and graph self-supervised learning, demonstrating the improvements over baselines and the advantages when modeling with mixed-curvature space.

Supplementary material includes a table of notation, a detailed discussion on the properties of our Radon transform and MCTSW distance, implementation detail of MCTSW, and additional experimental details.

## 2 BACKGROUND

We begin by presenting the necessary preliminaries on mixed-curvature spaces and the family of Sliced Wasserstein distance.

### 2.1 CONSTANT-CURVATURE AND MIXED-CURVATURE SPACES

**Constant-Curvature Spaces.** A $d$-dimensional constant curvature space (CCS) $\mathcal{C}_K^d$ is a smooth Riemannian manifold with constant sectional curvature $K$. Depending on the sign of $K$, this gives rise to hyperbolic ($K < 0$), Euclidean ($K = 0$), or spherical ($K > 0$) geometries.

Embedded models for CCS in $\mathbb{R}^{d+1}$ include the Lorentz model (hyperboloid) for hyperbolic space,

$$\mathbb{H}_d^K = \left\{ (x_1, \ldots, x_{d+1}) \in \mathbb{R}^{d+1} : -x_1^2 + x_2^2 + \cdots + x_{d+1}^2 = \frac{1}{K}, \; x_1 > 0 \right\}, \tag{1}$$

and the sphere for spherical space,

$$\mathbb{S}_d^K = \left\{ (x_1, \ldots, x_{d+1}) \in \mathbb{R}^{d+1} : x_1^2 + x_2^2 + \cdots + x_{d+1}^2 = \frac{1}{K} \right\}. \tag{2}$$

At each point $u \in \mathcal{C}_K^d$, the tangent space $T_u\mathcal{C}_K^d$ is Euclidean, and the exponential and logarithmic maps $\exp_u^K : T_u\mathcal{C}_K^d \to \mathcal{C}_K^d$ and $\log_u^K : \mathcal{C}_K^d \to T_u\mathcal{C}_K^d$ provide diffeomorphisms that link the manifold with its tangent spaces.

Following (Skopek et al., 2020; Gao et al., 2023; Ganea et al., 2018), we employ stereographic projection to present a unified representation of these geometries. For $(\eta, x^\top) \in \mathbb{R}^{d+1}$, the projection and its inverse are

$$\rho_K\big((\eta, x^\top)^\top\big) = \frac{x}{1 + \sqrt{|K|}\,\eta}, \quad \rho_K^{-1}(y) = \left( \frac{1}{\sqrt{|K|}} \frac{1 - K\|y\|_2^2}{1 + K\|y\|_2^2}, \; \frac{2y^\top}{1 + K\|y\|_2^2} \right)^\top. \tag{3}$$

Geometrically, this maps either the sphere or the hyperboloid onto the hyperplane $x_1 = 0$ via projection from $(-1, 0, \ldots, 0)$. The induced models are the Poincaré ball $\mathbb{P}_d^K$ ($K < 0$) and the

projected hypersphere $\mathbb{D}_d^K$ ($K > 0$) in $\mathbb{R}^d$. Both are conformal to Euclidean space, with conformal factor $\lambda_x^K = 2/(1 + K\|x\|_2^2)$.

To define operators that could work with both $\mathbb{P}_d^K$, $\mathbb{D}_d^K$, and $\mathbb{R}^d$, we adopt the *gyrovector space* formalism (Ungar, 2008), presented in Appendix A. This framework provides a smooth extension of Euclidean operations, recovering standard structures in the flat limit (Skopek et al., 2020).

**Wrapped Normal Distribution.** A fundamental Gaussian-like distribution on these manifolds is the Wrapped Normal Distribution (WND) (Nagano et al., 2019; Skopek et al., 2020). Sampling proceeds as

$$v \sim \mathcal{N}(0, \Sigma) \in T_0\mathcal{M}, \quad u = \mathrm{PT}_{0 \to \mu_0}^{\mathcal{M}}(v) \in T_{\mu_0}\mathcal{M}, \quad z = \exp_{\mu_0}^{\mathcal{M}}(u). \tag{4}$$

**Mixed-Curvature Spaces.** A Mixed-Curvature Space (MCS) is the Cartesian product

$$\mathcal{M} = \prod_{j=1}^m \mathcal{C}_{K_j}^{d_j}, \tag{5}$$

where each component $\mathcal{C}_{K_j}^{d_j}$ is a constant curvature space with curvature $K_j$.

For a point $u = (u_1, \ldots, u_m) \in \mathcal{M}$, tangent spaces and maps are defined componentwise:

$$\begin{aligned}
T_u\mathcal{M} &= \prod_{j=1}^m T_{u_j}\mathcal{C}_{K_j}^{d_j}, \\
\mathrm{Exp}_u^{\mathcal{M}}(q) &= \left(\exp_{u_1}^{K_1}(q_1), \ldots, \exp_{u_m}^{K_m}(q_m)\right), \\
\mathrm{Log}_u^{\mathcal{M}}(x) &= \left(\log_{u_1}^{K_1}(x_1), \ldots, \log_{u_m}^{K_m}(x_m)\right).
\end{aligned} \tag{6}$$

The distance decomposes as $d_{\mathcal{M}}(x, y)^2 = \sum_{i=1}^m d_{K_i}(x_i, y_i)^2$, and the canonical measure factorizes as $\mu_{\mathcal{M}}(A_1 \times \cdots \times A_m) = \prod_{j=1}^m \mu_{\mathcal{C}_{K_j}^{d_j}}(A_j)$.

Operations such as Möbius addition and scalar multiplication are similarly componentwise extended. This construction has proven valuable in representation learning (Gu et al., 2019), generative modeling (Skopek et al., 2020), and continual learning (Gao et al., 2023).

## 2.2 THE SLICED VARIANTS OF WASSERSTEIN DISTANCE

**Sliced Wasserstein Distance.** Sliced Wasserstein distance define on the Radon transform (Helgason, 2011) $R : L^1(\mathbb{R}^d) \to L^1(\mathbb{R} \times \mathbb{S}^{d-1})$ such that

$$Rf(x, \theta) = \int_{\mathbb{R}^d} f(z)\,\delta\big(x - \langle z, \theta \rangle\big)\,dz, \tag{7}$$

where $\delta$ denotes the Dirac delta function and $\theta \in \mathbb{S}^{d-1}$. For $\mu, \nu \in \mathcal{P}(\mathbb{R}^d)$ with densities $f_\mu, f_\nu$, the Sliced Wasserstein distance (SW) (Rabin et al., 2012) is given by

$$\mathrm{SW}(\mu, \nu) = \int_{\mathbb{S}^{d-1}} \mathrm{W}_1\big(Rf_\mu(\cdot, \theta), Rf_\nu(\cdot, \theta)\big)\,d\sigma(\theta), \tag{8}$$

where $\sigma$ is the uniform probability measure on $\mathbb{S}^{d-1}$.

A Monte Carlo estimations with sampled directions $\theta_i \sim \sigma(\theta)$ yields

$$\widehat{\mathrm{SW}}_L(\mu, \nu) = \frac{1}{L}\sum_{i=1}^L \mathrm{W}_1\big(Rf_\mu(\cdot, \theta_i), Rf_\nu(\cdot, \theta_i)\big). \tag{9}$$

Beyond Euclidean slicing, manifold-adapted variants leverage projections compatible with the underlying geometry. Notable extensions include spherical slicing (Bonet et al., 2023a; Tran et al., 2024a), hyperbolic slicing (Bonet et al., 2022a), and slicing in Cartan–Hadamard manifolds (Bonet

et al., 2025). Sampling strategies have also been refined for efficiency and variance reduction (Nguyen et al., 2024b).

**Tree-Sliced Wasserstein.** The Tree-Sliced Wasserstein (TSW) distance generalizes classical sliced Wasserstein methods by projecting measures onto tree metrics structures formed from connected line segments rather than single projections (Indyk & Thaper, 2003; Tran et al., 2025a). This involves partitioning measures via a splitting map and computing Wasserstein distances along the induced tree system, where closed-form solutions for $W_1$ are available. Unlike prior works that fix the metric space as a tree metric (Le et al., 2019; Lin et al., 2025; Indyk & Thaper, 2003; Takezawa et al., 2021), our approach follows the setting in (Tran et al., 2025a; 2024c), wherein the tree system is sampled dynamically. This design specifically accommodates measures with dynamic support, enabling greater flexibility and applicability.

## 3 TREE SYSTEMS IN MIXED-CURVATURE SPACES

In this section, we introduce the construction of tree systems within mixed-curvature spaces, which provides the geometric foundation for our generalized Sliced Wasserstein framework. We subsequently propose a Radon-type transform operator defined on such tree systems.

### 3.1 MIXED-CURVATURE TREE SYSTEMS

**Definition 3.1** (Geodesic Ray in $\mathcal{M}$). Let $\mathcal{M}$ be a mixed-curvature space. For $x, y \in \mathcal{M}$ with $y \neq x$, the geodesic ray $r_x^y$ originating at $x$ in direction $y$ is defined as

$$r_x^y := \bigsqcup_{t>0} \left\{ \mathrm{Exp}_x^{\mathcal{M}}\left(t \cdot \mathrm{Log}_x^{\mathcal{M}}(y)\right) \right\}. \tag{10}$$

**Remark 3.2.** Any pair $(t, r_x^y)$ with $t > 0$ uniquely corresponds to a point in $\mathcal{M}$. Thus, this parametric representation establishes a natural correspondence between points in the mixed-curvature space and elements of the tree system.

We now construct a tree structure generated by $k$ geodesic rays emanating from a common root $x \in \mathcal{M}$. Let $y_1, \ldots, y_k \in \mathcal{M}$ be $k$ points such that

$$d_{\mathcal{M}}(x, y_i) = \epsilon \quad \text{for all } i = 1, \ldots, k, \tag{11}$$

where $\epsilon > 0$ is a small constant. This normalization ensures that the directions of the rays are well-defined, thereby determining the family of rays $\{r_x^{y_i}\}_{i=1}^k$.

Consider the disjoint union $\bigsqcup_{i=1}^k r_x^{y_i}$ equipped with the equivalence relation $\sim$ defined by

$$(t_i, r_x^{y_i}) \sim (t_j, r_x^{y_j}) \iff t_i = t_j = 0. \tag{12}$$

We define the *mixed-curvature tree* as the quotient space

$$\mathcal{T}_x^{y_1, \ldots, y_k} := \bigsqcup_{i=1}^k r_x^{y_i} \Big/ \sim, \tag{13}$$

which has root $x$ and edges given by the rays $\{r_x^{y_i}\}_{i=1}^k$. A canonical measure on $\mathcal{T}_x^{y_1, \ldots, y_k}$ is induced by the standard Borel measure on $[0, \infty)$ through the parametric representation of geodesic rays.

We now endow $\mathcal{T}_x^{y_1, \ldots, y_k}$ with a tree metric.

**Definition 3.3** (Tree Metric on $\mathcal{T}$). For points $a = (t_i, r_x^{y_i})$ and $b = (t_j, r_x^{y_j})$ in $\mathcal{T}_x^{y_1, \ldots, y_k}$, the distance is defined as

$$d_{\mathcal{T}}(a, b) := \begin{cases} |t_i - t_j| & \text{if } i = j, \\ t_i + t_j & \text{if } i \neq j. \end{cases} \tag{14}$$

**Theorem 3.4** (Metric Structure of Mixed-Curvature Trees). *The space $(\mathcal{T}_x^{y_1, \ldots, y_k}, d_{\mathcal{T}})$ is a metric space endowed with a tree metric.*

The proof follows directly from standard arguments for tree metrics, as in (Tran et al., 2024c), verifying non-negativity, symmetry, the triangle inequality, and separation of points.

**Remark 3.5** (Geometric Interpretation). The metric $d_{\mathcal{T}}$ encodes the tree structure within mixed-curvature geometry. Specifically, when two points lie on the same geodesic ray ($i = j$), their distance coincides with the ambient metric along that ray. Conversely, when two points lie on different rays ($i \neq j$), their distance is computed by traveling through the common root $x$, which links rays that may correspond to regions of differing curvature.

### 3.2 RADON TRANSFORM ON MIXED-CURVATURE TREES

To formally define the Radon transform, we first need to specify how integration is defined on trees. For a tree $\mathcal{T}_x^{y_1,\dots,y_k}$, the integral of a function $f$ is given by

$$\int_{\mathcal{T}} f(z)\, d\sigma(z) := \sum_{i=1}^{k} \int_0^{\infty} f(t, r_x^{y_i})\, dt. \tag{15}$$

The associated space $L^1(\mathcal{T})$ is then understood as the collection of all real-valued functions on $\mathcal{T}$ whose absolute value admits a finite integral with respect to this measure.

To define the Radon transform onto a mixed-curvature tree, two components are required:

1. A *projection function* that assigns each point $z \in \mathcal{M}$ a coordinate on the rays of $\mathcal{T}$.

2. A *splitting map* $\alpha : \mathcal{M} \to \Delta_{k-1}$, where $\Delta_{k-1}$ is the $(k-1)$-dimensional probability simplex, specifying how the mass of each point is distributed across the rays.

For the projection function, we assign each point $z$ to the coordinate that equal to $d_{\mathcal{M}}(z, x)$, the geodesic distance to root, for every ray. We define the splitting map via the softmax of geodesic distances:

$$\alpha(z, \mathcal{T})_i = \frac{\exp(-d_{\mathcal{M}}(z, r_x^{y_i}))}{\sum_{j=1}^{k} \exp(-d_{\mathcal{M}}(z, r_x^{y_j}))}, \tag{16}$$

where $d_{\mathcal{M}}(z, r_x^{y_i}) = \min\limits_{h \in \bar{r}_x^{y_i}} d_{\mathcal{M}}(h, z)$, and $\bar{r}_x^{y_i}$ denotes the complete geodesic line through $x$ and $y_i$.

Other design choices for projection function and splitting map are discussed in Appendix F.

**Definition 3.6** (Radon Transform on Mixed-Curvature Trees). Let $\mathbb{T}_k^{\mathcal{M}}$ denote the collection of all mixed-curvature trees with $k$ rays in $\mathcal{M}$. For $\alpha \in C(\mathcal{M} \times \mathbb{T}_k^{\mathcal{M}}, \Delta_{k-1})$, the Radon transform is the operator

$$\mathcal{R}^{\alpha} : L^1(\mathcal{M}) \longrightarrow \prod_{\mathcal{T} \in \mathbb{T}_k^{\mathcal{M}}} L^1(\mathcal{T}) \tag{17}$$

defined by

$$\left(\mathcal{R}^{\alpha} f\right)_{\mathcal{T}}(t, r_x^{y_i}) = \int_{\mathcal{M}} f(z)\, \alpha(z, \mathcal{T})_i\, \delta\big(t - d_{\mathcal{M}}(x, z)\big)\, d\sigma(z), \tag{18}$$

for each $\mathcal{T} \in \mathbb{T}_k^{\mathcal{M}}$, where $\delta$ is the Dirac delta distribution.

A detailed discussion of the properties of this operator, including well-definedness, linearity, and injectivity, is provided in Appendix B.

## 4 MIXED CURVATURE TREE SLICED WASSERSTEIN DISTANCE

We are ready to present our novel distributional distance on Mixed-Curvature space in this section.

**Definition 4.1** (Mixed Curvature Tree Sliced Wasserstein Distance). The Mixed Curvature Tree Sliced Wasserstein distance between two probability measures $\nu, \mu \in \mathcal{P}(\mathcal{M})$ is defined as

$$\mathrm{MCTSW}(\nu, \mu) = \int_{\mathbb{T}_k^{\mathcal{M}}} \mathrm{W}_{d_{\mathcal{T}}}\big(\mathcal{R}_{\mathcal{T}}^{\alpha} \nu, \mathcal{R}_{\mathcal{T}}^{\alpha} \mu\big)\, d\sigma(\mathcal{T}), \tag{19}$$

where $\sigma$ is a probability measure on the space $\mathbb{T}_k^{\mathcal{M}}$ of mixed curvature trees with $k$ edges, and $W_{d_{\mathcal{T}}, 1}$ denotes the 1-Wasserstein distance on the tree metric space $(\mathcal{T}, d_{\mathcal{T}})$.

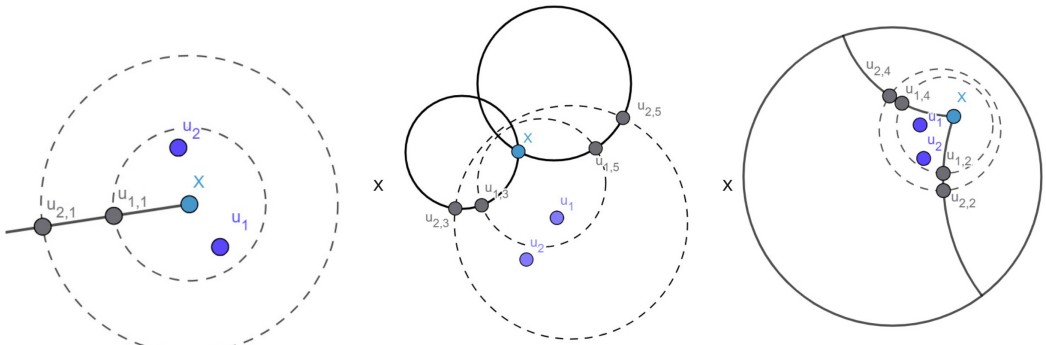

Figure 2: Illustration of the tree system (in bold) and the Radon transform onto the mixed-curvature tree in $\mathbb{E}_2 \times \mathbb{D}_2 \times \mathbb{P}_2$. The tree system consists of 5 rays, where each ray is represented by a ray in one component and a point in the others. The Radon transform projects the points $u_1$ and $u_2$ onto the $i^{\text{th}}$ geodesic at $u_{1,i}$ and $u_{2,i}$, respectively.

The metricity of MCTSW followed by the definition and the injectivity of our Radon transform which we defered the discussion to Appendix C.

In practice, the integral in Definition 4.1 is approximated by Monte Carlo estimator:

$$\widehat{\text{MCTSW}}_L(\nu, \mu) = \frac{1}{L} \sum_{i=1}^{L} \mathrm{W}_{d_{\mathcal{T}_i}}\left(\mathcal{R}_{\mathcal{T}_i}^{\alpha} \nu, \mathcal{R}_{\mathcal{T}_i}^{\alpha} \mu\right), \tag{20}$$

where $\{\mathcal{T}_i\}_{i=1}^{L}$ are independently sampled from the distribution $\sigma$ on $\mathbb{T}_k^{\mathcal{M}}$.

**Closed-form formula of $W_{d_{\mathcal{T}_i,1}}$.** Let $\mathcal{T}$ be a rooted tree (as a graph) with non-negative edge lengths, and the ground metric $d_{\mathcal{T}}$, i.e., the length of the unique path between two nodes. Given two probability distributions $\mu$ and $\nu$ supported on nodes of $\mathcal{T}$, the Wasserstein distance with ground metric $d_{\mathcal{T}}$ has closed form as follows:

$$W_{d_{\mathcal{T}},1}(\mu, \nu) = \sum_{e \in \mathcal{T}} w_e \cdot |\mu(\Gamma(v_e)) - \nu(\Gamma(v_e))|, \tag{21}$$

where $v_e$ is the endpoint of edge $e$ that is farther away from the tree root, $\Gamma(v_e)$ is the subtree of $\mathcal{T}$ rooted at $v_e$, and $w_e$ is the length of $e$. The derivation detail is provided in Appendix D.2.

**Simplifications and Tree Sampling.** To enable parallel computation and simplify the model, we impose the following constraints:

1. Each edge $y_i$ differs from the root $x$ in exactly one component:

$$\exists i : (y_i - x)^i \neq 0, \quad \text{and} \quad (y_i - x)^j = 0, \quad \forall j \neq i,$$

2. All components have equal dimension $d_i = d$.

Under these assumptions, the space $\mathbb{T}_k^{\mathcal{M}}$ of mixed curvature trees is in bijection with

$$\mathcal{M} \times [m]^k \times (\mathcal{M}_\epsilon)^k,$$

where $\mathcal{M}_\epsilon \subset \mathbb{R}^d$ denotes the $d$-dimensional sphere of radius $\epsilon$. This identification endows $\mathbb{T}_k^{\mathcal{M}}$ with a natural measure inherited from the product measure. Sampling proceeds by:

1. Sampling the root $x \in \mathcal{M}$ according to a Wrapped Normal distribution,

2. Sampling the components for the $k$ edges from $[m]^k$,

3. Sampling directions $(\ominus x \oplus y_1), \ldots, (\ominus x \oplus y_k) \in \mathcal{M}_\epsilon$. Alternatively, directions could be sampled on the sphere $\mathbb{S}^d$ since each $y_i$ is only required for computing the distance to the ray, and only the direction in the ambient space is needed.

**Time Complexity.** The time complexity of the algorithm is $\mathcal{O}(Ln \log n + Ldmnk)$, where $L$ is the number of trees used for Monte Carlo estimation, $k$ denotes the number of rays per tree, $n$ is the number of support points, $d$ denotes the dimension of each component, and $m$ is the number of components.

The first term, $\mathcal{O}(Ln \log n)$, arises from the sorting step, which is performed once per tree. Because all coordinates along each ray are identical after projection onto the root, a single sorting operation suffices for the entire tree. The second term, $\mathcal{O}(Ldmnk)$, accounts for the bulk of the mixed-curvature operators. Moreover, MCTSW is naturally well-suited to parallelization, since most computations are independent across components, which allows efficient implementation on GPUs and distributed systems. The overall algorithm for MCTSW is detailed in Algorithm 1.

---

**Algorithm 1:** Mixed Curvature Tree Sliced Wasserstein distance

**Input:** $\mathcal{M} = \times_{i=1}^{m} \mathcal{C}_i^d, \nu, \mu \in \mathcal{P}(\mathcal{M})$ as $\nu(x) = \sum_{j=1}^{n} u_j \, \delta(x - a_j)$ and $\mu(x) = \sum_{j=1}^{n} v_j \, \delta(x - a_j)$; number of trees $L$; number of rays per tree $k$; splitting maps $\alpha$.

**for** $\ell = 1$ **to** $L$ **do**

    Sample $x^{(\ell)} \sim \mathcal{WN}(0, \mathbb{I}_d)$;

    $\hat{y}_i^{(\ell)} \sim Unif(\mathbb{S}^d)$;

    $index_i^{(\ell)} \sim Unif[1, m]$;

    $\tilde{y}_i^{(\ell)} = \begin{cases} x^j + (\hat{y}_i^{(\ell)})^j & \text{if } j = index_i^{(\ell)} \\ x^j & \text{otherwise} \end{cases}$

    $y_i^{(\ell)} = M^x \cap r_x^{\tilde{y}_i^{(\ell)}}$;

    Construct the mixed curvature tree $\mathcal{T}_\ell = \mathcal{T}_{y_1^{(\ell)}, \ldots, y_k^{(\ell)}}^{x^{(\ell)}}$;

    Compute $W_{d_{\mathcal{T}_\ell}, 1}(\mathcal{R}_{\mathcal{T}_\ell}^\alpha \nu, \mathcal{R}_{\mathcal{T}_\ell}^\alpha \mu)$ by Eq **??**;

Compute $\widehat{\mathrm{MCTSW}}(\nu, \mu) = \dfrac{1}{L} \sum_{\ell=1}^{L} W_{d_{\mathcal{T}_\ell}}(\mathcal{R}_{\mathcal{T}_\ell}^\alpha \nu, \mathcal{R}_{\mathcal{T}_\ell}^\alpha \mu)$;

**return** $\widehat{\mathrm{MCTSW}}(\nu, \mu)$

---

## 5 EXPERIMENTS

We present experiments to demonstrate: i. the effectiveness of our proposed Mixed-Curvature Tree-Sliced Wasserstein (MCTSW) distance, and ii. the advantages of modeling data in mixed-curvature spaces compared to constant-curvature alternatives. We evaluate across three settings: gradient flow, variational autoencoders (VAEs), and graph self-supervised learning. Experimental details and training time for these experiments are provided in Appendix E.

### 5.1 GRADIENT FLOW

Our first experiment considers learning a target distribution $\nu$ from a source distribution $\mu$ using Riemannian gradient descent on $\mathrm{MCTSW}(\nu, \mu)$. The target distribution $\nu$ is constructed from 2,400 samples of a mixture of six wrapped-normal distributions (WNDs) arranged equidistantly from the origin, while the source $\mu$ is a single WND centered at the origin.

We compare MCTSW against three baselines: i. $\mathrm{SW}_{\text{ambient}}$: sliced Wasserstein (Bonneel et al., 2015) computed in the ambient space; ii. Prod-TSW: product metric applying Tree-Sliced Wasserstein independently on each factor space; iii. Prod-SW: product metric applying geodesic sliced Wasserstein in each factor space (Rabin et al., 2012; Bonet et al., 2023b;a).

All methods are trained with full-batch updates for 10,000 iterations, using 1,800 random projections per run. Table 1 shows that MCTSW achieves a substantially lower geodesic $\log W_2$ than other baselines. This suggests improved convergence behavior and more faithful recovery of the target distribution in mixed-curvature spaces.

## 5.2 GRAPH SELF-SUPERVISED LEARNING

We further benchmark MCTSW in a graph self-supervised learning (SSL) setting, following (Liu et al., 2024). Specifically, we learn node representations using a loss function composed of an alignment term and a uniformity term:

$$\mathcal{L} = \mathcal{L}_{ali} + \lambda \mathcal{L}_{uni},$$

where the uniformity loss is conventionally expressed as the 2-Wasserstein distance between hidden representations and a Gaussian prior. In our approach, we replace this term with MCTSW when modeling the latent space as an MCS manifold, obtaining MCTSW-SSGE, and compare against constant-curvature variants (E-, H-, and S-TSW-SSGE), which model the latent space in Euclidean, Hyperbolic, and Spherical spaces, respectively, while using MCTSW restricted to a single component as uniformity loss.

Table 1: Learning a target mixture of 6 WNDs.

| Method | $\log W_2$ ($\downarrow$) |
| --- | --- |
| SW$_{\text{ambient}}$ | 0.33 |
| Prod-TSW | 0.34 |
| Prod-SW | 0.31 |
| MCTSW (ours) | $-\mathbf{3.65}$ |

Table 2: Test Accuracy ($\uparrow$) on Cora.

| Method | Accuracy ($\uparrow$) |
| --- | --- |
| SSGE | $79.55 \pm 0.35$ |
| E-TSW-SSGE | $77.85 \pm 0.32$ |
| H-TSW-SSGE | $75.10 \pm 0.22$ |
| S-TSW-SSGE | $78.33 \pm 0.15$ |
| MCTSW-SSGE | $\mathbf{79.86 \pm 0.45}$ |

Obtaining the representation, we evaluate its quality through downstream classification accuracy using a random forest classifier on MCS (Chlenski et al., 2025). The results, shown in Table 2, demonstrate slight improvements over the baseline SSGE (Liu et al., 2024) on the Cora dataset (Kipf & Welling, 2017) when using MCTSW as the uniformity loss. Furthermore, modeling with an MCS latent space yields higher accuracy compared to constant-curvature variants.

## 5.3 MIXED-CURVATURE VARIATIONAL AUTOENCODERS

Variational Autoencoders (VAEs) (Kingma & Welling, 2013) are probabilistic generative models that learn continuous latent representations, enabling both faithful reconstructions and realistic data generation via sampling. Standard VAEs optimize a reconstruction loss combined with Kullback–Leibler (KL) divergence regularization, but subsequent work has investigated alternative divergences such as Wasserstein distances (Tolstikhin et al., 2018; Kolouri et al., 2018a), as well as non-Euclidean latent geometries including spherical (Davidson et al., 2018), hyperbolic (Bonet et al., 2022a), and mixed-curvature spaces (MCS) (Skopek et al., 2020).

We evaluate our proposed MCTSW regularizer within a mixed-curvature latent space on CIFAR-10 (Krizhevsky & Hinton, 2009), optimizing a binary cross-entropy (BCE) reconstruction objective. We compare against a range of VAE baselines: the vanilla Euclidean VAE with KL divergence (Kingma & Welling, 2013); SWAE (Kolouri et al., 2018a) (Euclidean latent space with Sliced Wasserstein); S-VAE (Davidson et al., 2018) (spherical latent space with KL divergence); STSW-VAE (Tran et al., 2025c) (spherical latent space with tree-sliced Wasserstein); H-VAE (hyperbolic latent space with KL divergence); HSW-VAE (Bonet et al., 2022a) (hyperbolic latent space with hyperbolic SW); and M-VAE (Skopek et al., 2020) (MCS latent space with KL divergence).

As shown in Table 3, incorporating MCTSW within MCS improves performance compared to i. the KL-based mixed-curvature baseline M-VAE, demonstrating the superiority of MCTSW over KL divergence for regularization, and ii. constant-curvature KL- and SW-based variants, highlighting the advantages of modeling with MCS when combined with the MCTSW distance.

## 6 CONCLUSION

This paper introduced the Mixed-Curvature Tree-Sliced Wasserstein Distance (MCTSW), a novel measure leveraging tree systems to accurately represent both geometric and topological structures

Table 3: Test BCE (↓) on CIFAR10.

| Latent space | Method | Regularizer | Test BCE (↓) |
|---|---|---|---|
| Euclidean | VAE | KL-divergence | $0.6423 \pm 0.0008$ |
| | SWAE | SW | $0.6043 \pm 0.0005$ |
| Spherical | S-VAE | KL-divergence | $0.6285 \pm 0.0004$ |
| | STSW-VAE | STSW | $0.6026 \pm 0.0009$ |
| Hyperbolic | H-VAE | KL-divergence | $0.6402 \pm 0.0005$ |
| | HSW-VAE | HSW | $0.6012 \pm 0.0006$ |
| MCS | M-VAE | KL-divergence | $0.6419 \pm 0.0008$ |
| | MCTSW-VAE | MCTSW | $\mathbf{0.6000 \pm 0.0002}$ |

in mixed-curvature spaces. By adapting Radon transforms to tree-based representations, MCTSW offers a tractable and closed-form method for comparing probability distributions. Our theoretical analysis demonstrates key properties of the Radon transform and the MCTSW distance, ensuring the reliability for distributional comparisons. Empirical results in three main tasks: gradient flow, variational auto-encoder, and graph self-supervised learning, show the advantages of MCTSW over baselines methods and also illustrate the benefit of mixed-curvature latent representations over constant-curvature alternatives for modeling complex datasets. The method's design allows for efficient parallel computation, enhancing scalability; however, the computational overhead from gyrovector space operations poses challenges for runtime and numerical stability, which warrant further investigation.

ACKNOWLEDGMENTS

This research / project is supported by the National Research Foundation Singapore under the AI Singapore Programme (AISG Award No: AISG2-TC-2023-012-SGIL). This research / project is supported by the Ministry of Education, Singapore, under the Academic Research Fund Tier 1 (FY2023) (A-8002040-00-00, A-8002039-00-00). This research / project is also supported by the NUS Presidential Young Professorship Award (A-0009807-01-00) and the NUS Artificial Intelligence Institute–Seed Funding (A-8003062-00-00).

**Ethics Statement.** Given the nature of the work, we do not foresee any negative societal and ethical impacts of our work.

**Reproducibility Statement.** Source codes for our experiments are provided in the supplementary materials of the paper. The details of our experimental settings and computational infrastructure are given in Section 5 and Appendix E. All datasets that we used in the paper are published, and they are easy to access in the Internet.

**LLM Usage Declaration.** We use large language models (LLMs) for grammar checking and correction.

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

## TABLE OF NOTATION

| | |
|---|---|
| $\mathbb{R}^d$ | $d$-dimensional Euclidean space |
| $\|\cdot\|_2$ | Euclidean norm |
| $\mathbb{S}^{d-1}$ | $(d-1)$-dimensional hypersphere |
| $\sqcup$ | disjoint union |
| $L^1(X)$ | space of Lebesgue integrable functions on $X$ |
| $\mathcal{P}(X)$ | space of probability measures on $X$ |
| $\sigma(\cdot)$ | distribution on a space |
| $\mu, \nu$ | measures |
| $\delta(\cdot)$ | 1-dimensional Dirac delta function |
| $\mathcal{U}(\cdot)$ | uniform distribution on a sample space |
| $[n, m]$ | set of integers from $n$ to $m$ |
| $\mathcal{C}(X, Y)$ | space of continuous maps from $X$ to $Y$ |
| $d(\cdot, \cdot)$ | metric in metric space |
| $d_{\mathcal{T}}(\cdot, \cdot)$ | tree metric |
| $\Lambda$ | (rooted) subtree |
| $\mathcal{T}$ | tree system |
| $L$ | number of Monte Carlo samples |
| $k$ | number of lines in a system of lines or a tree system |
| $\mathcal{R}^{\alpha}$ | Radon Transform on Systems of Lines |
| $\Delta_{k-1}$ | $(k-1)$-dimensional standard simplex |
| $\alpha$ | splitting map |
| $\mathbb{T}$ | space of tree systems |
| $\mathcal{N}$ | normal (Gaussian) distribution |
| $\mathcal{U}$ | uniform distribution |
| $\delta$ | Dirac delta distribution |

## A  GYROVECTORSPACE

In this section, we summarize some important operations in gyrovectorspaces.

**Curved Trigonometry.**

$$
\sin_K(t) = \begin{cases} \frac{\sinh(t\sqrt{-K})}{\sqrt{-K}}, & K < 0, \\ t, & K = 0, \\ \frac{\sin(t\sqrt{K})}{\sqrt{K}}, & K > 0, \end{cases} \quad \sin_K^{-1}(t) = \begin{cases} \frac{\operatorname{arsinh}(t\sqrt{-K})}{\sqrt{-K}}, & K < 0, \\ t, & K = 0, \\ \frac{\arcsin(t\sqrt{K})}{\sqrt{K}}, & K > 0, \end{cases} \quad (22)
$$

$$
\tan_K(t) = \begin{cases} \tanh(t), & K < 0 \\ t, & K = 0 \\ \tan(t), & K > 0 \end{cases} \quad \tan_K^{-1}(t) = \begin{cases} \operatorname{artanh}(t), & K < 0 \\ t, & K = 0 \\ \arctan(t), & K > 0 \end{cases} \quad (23)
$$

**Möbius Addition**  (analogous to vector addition in vector space).

$$
x \oplus_K y = \frac{(1 - 2K\langle x, y \rangle - K\|y\|_2^2)x + (1 + K\|x\|_2^2)y}{1 - 2K\langle x, y \rangle + K^2\|x\|_2^2\|y\|_2^2}, \tag{24}
$$

**Scalar Multiplication**  (analogous to vector-scalar multiplication in vector space).

$$
t \otimes_K x = \begin{cases} \frac{\tan_K\left(t\,\tan_K^{-1}(\sqrt{|K|}\,\|x\|_2)\right)}{\sqrt{|K|}\,\|x\|_2}\, x, & K \neq 0,\ x \neq 0, \\ 0, & K \neq 0,\ x = 0, \\ t\,x, & K = 0, \end{cases} \tag{25}
$$

**Gyration.**

$$
\operatorname{gyr}[x, y]\, v = \ominus_K(x \oplus_K y)\ \oplus_K\ (x \oplus_K (y \oplus_K v)), \tag{26}
$$

where $\ominus_K x = -x$.

**Exponential Map**  (mapping from tagent space to manifold).

$$
\exp_u^K(x) = \begin{cases} u \oplus_K \left( \frac{\tan_K(\sqrt{|K|}\,\lambda_u^K\|x\|_2/2)}{\sqrt{|K|}\,\|x\|_2}\, x \right), & K \neq 0,\ x \neq 0, \\ u, & K \neq 0,\ x = 0, \\ u + x, & K = 0, \end{cases} \tag{27}
$$

**Logarithm Map**  (inversion of exponential map).

$$
\log_u^K(x) = \begin{cases} \frac{2}{\sqrt{|K|}\,\lambda_u^K}\,\tan_K^{-1}\left(\sqrt{|K|}\,\|(-u)\oplus_K x\|_2\right)\frac{(-u)\oplus_K x}{\|(-u)\oplus_K x\|_2}, & \text{if } K \neq 0,\ u \neq x, \\ 0, & \text{if } K \neq 0,\ u = x, \\ -u + x, & \text{if } K = 0, \end{cases} \tag{28}
$$

**Gyrodistance.**

$$
d_K(x, y) = \begin{cases} \frac{2}{\sqrt{|K|}}\,\tan_K^{-1}\left(\sqrt{|K|}\,\|(-x)\oplus_K y\|_2\right), & K \neq 0, \\ 2\|x - y\|_2, & K = 0, \end{cases} \tag{29}
$$

**Parallel Transport.**  Along the geodesic from $x$ to $y$:

$$
\operatorname{PT}_{x \to y}^K(v) = \frac{\lambda_x^K}{\lambda_y^K}\,\operatorname{gyr}[y, -x]\, v, \tag{30}
$$

**Geodesics.**  (analogous to lines in vectorspace) The geodesic connecting $x$ and $y$ is parameterized as

$$
\varphi(t) = \exp_x^K\left(t\,\log_x^K(y)\right) = x \oplus_K t \otimes_K ((-x) \oplus_K y), \quad t \in \mathbb{R}. \tag{31}
$$

# B  BASIC PROPERTIES OF RADON TRANSFORM ONTO MIXED-CURVATURE TREE.

We establish the basic properties of our Radon transform, including the well-definedness, linearity, and injectivity.

**Theorem B.1** (Well-definedness of $\mathcal{R}_\mathcal{T}^\alpha$). *The operator $\mathcal{R}_\mathcal{T}^\alpha$ is well-defined.*

*Proof.* We compute:

$$\|\mathcal{R}_\mathcal{T}^\alpha f\|_\mathcal{T} = \sum_{i=1}^k \int_0^\infty \left| \int_\mathcal{M} f(z)\, \alpha(z,\mathcal{T})_i\, \delta\big(t - d_\mathcal{M}(x,z)\big)\, d\sigma(z) \right| dt$$

$$\leq \sum_{i=1}^k \int_0^\infty \int_\mathcal{M} |f(z)|\, \alpha(z,\mathcal{T})_i\, \delta\big(t - d_\mathcal{M}(x,z)\big)\, d\sigma(z)\, dt$$

$$= \sum_{i=1}^k \int_\mathcal{M} |f(z)|\, \alpha(z,\mathcal{T})_i \left( \int_0^\infty \delta\big(t - d_\mathcal{M}(x,z)\big)\, dt \right) d\sigma(z) \tag{32}$$

$$= \sum_{i=1}^k \int_\mathcal{M} |f(z)|\, \alpha(z,\mathcal{T})_i\, d\sigma(z)$$

$$= \int_\mathcal{M} |f(z)| \left( \sum_{i=1}^k \alpha(z,\mathcal{T})_i \right) d\sigma(z) \tag{33}$$

$$= \int_\mathcal{M} |f(z)|\, d\sigma(z)$$

$$= \|f\|_1 \tag{34}$$

$$< \infty. \tag{35}$$

The interchange of integration order in Eq 32 and Eq 33 is justified by Fubini's theorem, as the integrand is non-negative and the final integral is finite. $\square$

**Theorem B.2** (Linearity of $\mathcal{R}_\mathcal{T}^\alpha$). *The operator $\mathcal{R}_\mathcal{T}^\alpha$ is linear, that is,*

$$\mathcal{R}_\mathcal{T}^\alpha(af + bg) = a\mathcal{R}_\mathcal{T}^\alpha f + b\mathcal{R}_\mathcal{T}^\alpha g$$

*for all $f, g \in L^1(\mathcal{M})$ and $a, b \in \mathbb{R}$.*

*Proof.* For arbitrary $a, b \in \mathbb{R}$ and $f, g \in L^1(\mathcal{M})$, we have:

$$\mathcal{R}_\mathcal{T}^\alpha(af + bg)(t, r_x^{y_i})$$

$$= \int_\mathcal{M} (af(z) + bg(z))\, \alpha(z,\mathcal{T})_i\, \delta\big(t - d_\mathcal{M}(x,z)\big)\, d\sigma(z)$$

$$= a \int_\mathcal{M} f(z)\, \alpha(z,\mathcal{T})_i\, \delta\big(t - d_\mathcal{M}(x,z)\big)\, d\sigma(z)$$

$$\quad + b \int_\mathcal{M} g(z)\, \alpha(z,\mathcal{T})_i\, \delta\big(t - d_\mathcal{M}(x,z)\big)\, d\sigma(z)$$

$$= a\mathcal{R}_\mathcal{T}^\alpha f(t, r_x^{y_i}) + b\mathcal{R}_\mathcal{T}^\alpha g(t, r_x^{y_i}). \tag{36}$$

This confirms the linearity of the operator. $\square$

**Theorem B.3.** *The operator $\mathcal{R}^\alpha$ is injective.*

*Proof.* We begin by considering the case of single-ray trees. In this setting, the Radon transform reduces to the spherical mean transform in mixed-curvature space. This type of transform has been extensively studied in the context of constant-curvature spaces (Agranovsky et al., 2007; Nguyen, 2016). In particular, (Antipov et al., 2011) provide an explicit inversion formula for constant-curvature spaces, which directly establishes the injectivity property in that case. Moreover, on a

general real-analytic manifold, injectivity follows from the results of (Quinto, 2006), and therefore applies to our setting as a consequence.

We now proceed to the case of general trees in mixed-curvature space. Assume that

$$\mathcal{R}_{\mathcal{T}}^{\alpha} f = 0 \quad \forall \, \mathcal{T} \in \mathbb{T}_k^{\mathcal{M}}. \tag{37}$$

Consider the family of trees with $k$ coinciding rays, denoted by $\mathbb{T}_{k,r_x^y}$. For such trees, the coordinate system and the associated Radon-transformed mass are identical across all rays. Define the mapping

$$\mathcal{G} : \mathbb{T}_{r_x^y}^k \longrightarrow \mathbb{T}_{r_x^y}^1, \qquad \mathcal{G} f(t, r_x^y) = k f(t, r_x^y). \tag{38}$$

It is immediate that $\mathcal{G}$ induces a bijective correspondence between $\mathbb{T}_{r_x^y}^k$ and $\mathbb{T}_{r_x^y}^1$. Furthermore, we obtain

$$\mathcal{G} \, \mathcal{R}_{\mathcal{T}_k}^{\alpha} = \mathcal{R}_{\mathcal{T}_1}^{\alpha}. \tag{39}$$

Consequently, for every $\mathcal{T}_k \in \mathbb{T}_{k,r_x^y} \subset \mathbb{T}_k^{\mathcal{M}}$, we have

$$\mathcal{R}_{\mathcal{T}_1}^{\alpha} f = \mathcal{G} \, \mathcal{R}_{\mathcal{T}_k}^{\alpha} f = 0. \tag{40}$$

By the injectivity established in the case of single-ray trees, it follows that $f = 0$. Hence, we conclude that $\mathcal{R}^{\alpha}$ is injective. $\qquad \square$

## C   METRICITY OF MCSTSW

**Theorem C.1** (Metricity of Mixed Curvature Tree Sliced Wasserstein Distance)**.** *The Mixed Curvature Tree Sliced Wasserstein distance defines a metric on* $\mathcal{P}(\mathcal{M})$.

*Proof.* For the Mixed Curvature Tree Sliced Wasserstein distance being a metric, it must compile to three properties:

**Positive definitess.** As the tree distance $W_{d_{\mathcal{T}},1}(\nu, \mu) \geq \quad \forall \; \nu, \mu \; \in \; \mathcal{P}(\mathcal{T})$, it follows $\mathrm{MCTSW}(\nu, \mu) \geq 0$. Also, as $W_{d_{\mathcal{T}},1}(\nu, \nu) = 0 \; \forall \; \nu \in \mathcal{P}(\mathcal{T})$, $\mathrm{MCTSW}(\nu, \nu) = 0 \; \forall \; \nu \in \mathcal{P}(\mathcal{T})$. Positive definite follows by the injectivity of Radon transform.

**Symmetry.** For all $\nu, \mu \in \mathcal{P}(\mathcal{M})$, we have:

$$\mathrm{MCTSW}(\nu, \mu) = \int_{\mathbb{T}_k^{\mathcal{M}}} \mathrm{W}_{d_{\mathcal{T}}}(\mathcal{R}_{\mathcal{T}}^{\alpha} \nu, \mathcal{R}_{\mathcal{T}}^{\alpha} \mu) d\sigma(\mathcal{T})$$

$$= \int_{\mathbb{T}_k^{\mathcal{M}}} \mathrm{W}_{d_{\mathcal{T}}}(\mathcal{R}_{\mathcal{T}}^{\alpha} \mu, \mathcal{R}_{\mathcal{T}}^{\alpha} \nu) d\sigma(\mathcal{T})$$

$$= \mathrm{MCTSW}(\mu, \nu)$$

**Triangle inequality.**   For $\nu_1, \nu_2, \nu_3 \in \mathcal{P}(\mathbb{S}^d)$, we have

$$\mathrm{MCTSW}(\nu_1, \nu_2) + \mathrm{MCTSW}(\nu_2, \nu_3)$$

$$= \int_{\mathbb{T}_k^{\mathcal{M}}} \mathrm{W}_{d_{\mathcal{T}}}\left(\mathcal{R}_{\mathcal{T}}^{\alpha} \nu_1, \mathcal{R}_{\mathcal{T}}^{\alpha} \nu_2\right) d\sigma(\mathcal{T}) + \int_{\mathbb{T}_k^{\mathcal{M}}} \mathrm{W}_{d_{\mathcal{T}}}\left(\mathcal{R}_{\mathcal{T}}^{\alpha} \nu_2, \mathcal{R}_{\mathcal{T}}^{\alpha} \nu_3\right) d\sigma(\mathcal{T})$$

$$= \int_{\mathbb{T}_k^{\mathcal{M}}} \left(\mathrm{W}_{d_{\mathcal{T}}}\left(\mathcal{R}_{\mathcal{T}}^{\alpha} \nu_1, \mathcal{R}_{\mathcal{T}}^{\alpha} \nu_2\right) + \mathrm{W}_{d_{\mathcal{T}}}\left(\mathcal{R}_{\mathcal{T}}^{\alpha} \nu_2, \mathcal{R}_{\mathcal{T}}^{\alpha} \nu_3\right)\right) d\sigma(\mathcal{T})$$

$$\geq \int_{\mathbb{T}_k^{\mathcal{M}}} \mathrm{W}_{d_{\mathcal{T}}}\left(\mathcal{R}_{\mathcal{T}}^{\alpha} \nu_1, \mathcal{R}_{\mathcal{T}}^{\alpha} \nu_3\right) d\sigma(\mathcal{T})$$

$$= \mathrm{MCTSW}(\nu_1, \nu_3)$$

This confirm MCTSW is a metric on $\mathcal{P}(\mathcal{M})$. $\qquad \square$

# D  IMPLEMENTATION DETAILS

## D.1  COMPUTATION OF DISTANCE FROM A POINT TO A RAY

We first consider the single-component case, ie, $\mathcal{M} = \mathcal{C}_K^d$. If $z \in r_x^{y_i}$, then $d_{\mathcal{M}}(z, r_x^{y_i}) = 0$. Otherwise, applying the Generalized Law of Sines (Katok, 1992) gives, for any $h \in \overline{r_x^{y_i}}$,

$$
\begin{aligned}
\sin_K(d_{\mathcal{M}}(z, h)) &= \sin_K(d_{\mathcal{M}}(z, x)) \frac{\sin(\angle zxh)}{\sin(\angle zhx)} \\
&\geq \sin_K(d_{\mathcal{M}}(z, x)) \cdot \sin(\angle(xz, r_x^{y_i})),
\end{aligned}
\tag{41}
$$

where the inequality uses $0 < \sin(\cdot) \leq 1$.

The first term is computed via curved trigonometry; the angle in the second term is computed using left-Möbius addition as an isometry translating $(x, z, r_x^y)$ to $(0, \hat{z}, r_0^{\hat{y}})$ and preserving angles (Ungar, 2008). Thus,

$$
\cos(\angle(xy, xz)) = \frac{\langle y, z \rangle}{\|y\|_2 \|z\|_2}.
\tag{42}
$$

The distance is then computed with

$$
d_{\mathcal{M}}(z, r_x^y) = \sin_K^{-1} \left( \sin_K(d_{\mathcal{M}}(z, x)) \cdot \sin(\angle(xz, r_x^y)) \right).
\tag{43}
$$

For a product space assuming $(y - x)^i \neq 0$,

$$
d_{\mathcal{M}}(z, r_x^y)^2 = d_{\mathcal{C}_{K_i}^d}(z, r_x^y)^2 + \sum_{j \neq i} d_{\mathcal{C}_{K_j}^d}(z, x)^2.
\tag{44}
$$

## D.2  CLOSED-FORM OF WASSERSTEIN DISTANCE ON MIXED-CURVATURE TREE

In this subsection, we derive the closed-form of $W_{d_\tau, 1}(\mathcal{R}_{\mathcal{T}}^\alpha \nu, \mathcal{R}_{\mathcal{T}}^\alpha \mu)$.

Assume

$$
\nu(x) = \sum_{j=1}^n \nu_j \, \delta(x - a_j) \qquad \text{and} \qquad \mu(x) = \sum_{j=1}^n \mu_j \, \delta(x - a_j)
\tag{45}
$$

This representation unify the support of discrete distributions, allowing zero mass. Consider the mixed curvature tree $\mathcal{T} = \mathcal{T}_{y_1, \ldots, y_k}^x$. Let $c_j = d_{\mathcal{M}}(x, a_j) \ \forall \ j \in [1, n]$, and let $c_0 = 0$. Without loss the generality, we assume that

$$
0 = c_0 \leq c_1 \leq c_2 \leq \cdots \leq c_n
\tag{46}
$$

For $0 \leq j \leq n$ and $1 \leq i \leq k$, consider all points $x_j^{(i)} = (c_j, r_x^{y_i})$ on the mixed curvature tree $\mathcal{T}$. We have $x_0^{(1)} = x_0^{(2)} = \cdots = x_0^{(k)} = x$ and $x_j^{(i)}$ is the intersection between the ball $B_x = \{z \mid d_{\mathcal{M}}(z, x) = d_{\mathcal{M}}(x, a_j) = c_j\}$ and the ray $r_x^{y_i}$ forall $1 \leq j \leq n$. We compute $\mathcal{R}_{\mathcal{T}}^\alpha \nu(t, r_x^{y_i})$ for $t \in [0, +\infty)$ and $1 \leq i \leq k$,

$$
\begin{aligned}
\mathcal{R}_{\mathcal{T}}^\alpha \nu(t, r_x^{y_i}) &= \int_{\mathcal{M}} \nu(z) \, \alpha(z, \mathcal{T})_i \, \delta(t - d_{\mathcal{M}}(x, z)) \, d\sigma(z) \\
&= \int_{\mathcal{M}} \left( \sum_{j=1}^n \pi_j \, \delta(z - a_j) \right) \alpha(z, \mathcal{T})_i \, \delta(t - d_{\mathcal{M}}(x, z)) \, d\sigma(z) \\
&= \sum_{j=1}^n \nu_j \int_{\mathcal{M}} \alpha(z, \mathcal{T})_i \, \delta(z - a_j) \, \delta(t - d_{\mathcal{M}}(x, z)) \, d\sigma(z).
\end{aligned}
\tag{47}
$$

Hence,

$$
\mathcal{R}_{\mathcal{T}}^\alpha \nu(t, r_x^{y_i}) = \begin{cases} 0 & \text{if } t \notin \{c_1, \ldots, c_n\} \\ \alpha(a_j, \mathcal{T})_i \, \nu_j & \text{if } t = c_j \text{ for some } j \end{cases}
\tag{48}
$$

The formula for $\mathcal{R}^\alpha_\mathcal{T}\mu$ is achieved similarly.

For $1 \leq j \leq n$ and $1 \leq i \leq k$, let

$$\nu^{(i)}_j = \alpha(a_j, \mathcal{T})_i\, \nu_j, \qquad \mu^{(i)}_j = \alpha(a_j, \mathcal{T})_i\, \mu_j. \tag{49}$$

Consider $\mathcal{T}$ as a graph with nodes $x^{(i)}_j = (c_j, r^{y_i}_x)$ for $0 \leq j \leq n$, $1 \leq i \leq k$, where $x^{(i)}_0 = x$ for all $i$ (the root). The edges are

$$e^{(i)}_j = \big(x^{(i)}_{j-1}, x^{(i)}_j\big), \qquad 1 \leq i \leq k,\ 1 \leq j \leq n,$$

with length $\ell(e^{(i)}_j) = c_j - c_{j-1}$. For $j > 0$, the subtree $\Gamma\big(x^{(i)}_j\big)$ contains all nodes $x^{(i)}_p$ with $p \geq j$.

The Tree–Sliced Wasserstein distance on $\mathcal{T}$ is

$$W_{d_\tau}\big(\mathcal{R}^\alpha_\mathcal{T}\nu, \mathcal{R}^\alpha_\mathcal{T}\mu\big) = \sum_{i=1}^{k}\sum_{j=1}^{n}\big(c_j - c_{j-1}\big)\left|\nu\big(\Gamma(x^{(i)}_j)\big) - \mu\big(\Gamma(x^{(i)}_j)\big)\right|. \tag{50}$$

From

$$\nu\big(\Gamma(x^{(i)}_j)\big) = \sum_{p=j}^{n}\nu^{(i)}_p, \qquad \mu\big(\Gamma(x^{(i)}_j)\big) = \sum_{p=j}^{n}\mu^{(i)}_p,$$

we obtain

$$W_{d_\tau}\big(\mathcal{R}^\alpha_\mathcal{T}\nu, \mathcal{R}^\alpha_\mathcal{T}\mu\big) = \sum_{i=1}^{k}\sum_{j=1}^{n}\big(c_j - c_{j-1}\big)\left|\sum_{p=j}^{n}\big(\nu^{(i)}_p - \mu^{(i)}_p\big)\right| \tag{51}$$

$$= \sum_{i=1}^{k}\sum_{j=1}^{n}\big(c_j - c_{j-1}\big)\left|\sum_{p=j}^{n}\alpha(a_p, \mathcal{T})_i\,(\nu_p - \mu_p)\right|. \tag{52}$$

# E  EXPERIMENTAL DETAILS

## E.1  GRADIENT FLOW

Our gradient flow experiment aims at matching distributions on the product manifold $\mathbb{E}_2 \times \mathbb{P}_2 \times \mathbb{D}_2$. We use a mixture of 6 WNDs, each with $\Sigma = 0.01\mathbb{I}$, centered at distance $0.5$ from the origin and equally spaced (equiangular) between consecutive centers, as the target distribution $\nu$. The source distribution $\mu$ is a WND centered at the origin. We use full-batch Riemannian gradient descent (Boumal, 2023) to minimize $\mathrm{MCTSW}(\nu, \mu)$. Both the source and target consist of 2,400 data points. We tune the learning rate for each method while keeping the total number of projections fixed at 1,800.

The baselines are:

- $SW_{\text{ambient}}$: SW distance in ambient space $\mathbb{R}^6$
- Prod-TSW: 1-product metric using the MCTSW variant on each component space
- Prod-SW: 1-product metric using the geodesic SW in each component

## E.2  VARIATIONAL AUTO-ENCODER

**Baselines.** The vanilla VAE (Kingma & Welling, 2013) uses Euclidean space as latent space, with the objective consisting of Binary Cross Entropy as reconstruction loss and KL-divergence as regularizer. SWAE (Kolouri et al., 2018a) replaces the KL-divergence with Sliced-Wasserstein in Euclidean space. S-VAE (Davidson et al., 2018) uses the sphere as latent space and KL-divergence as regularizer. STSW-VAE replaces the KL-divergence on the sphere with the Sphere Tree-Sliced Wasserstein distance (Tran et al., 2025c). H-VAE uses hyperbolic space as latent space and KL-divergence as regularizer. HSW-VAE replaces the KL-divergence with the hyperbolic sliced Wasserstein distance (Bonet et al., 2022a). M-VAE uses MCS as latent space and KL-divergence as regularizer. The latent spaces for the Euclidean, Spherical, Hyperbolic, and MCS cases are $\mathbb{E}_9$, $\mathbb{S}_9$, $\mathbb{P}_9$,

and $\mathbb{E}_3 \times \mathbb{D}_3 \times \mathbb{P}_3$, respectively. All methods use a mixture of 10 WNDs (or vMF for the spherical case, and Normal distribution for the Euclidean case) as the prior distribution. For the MCS and Hyperbolic cases, after passing through the encoder, embeddings are mapped to the manifold using the exponential map. With this parameterization, we can utilize Euclidean optimizers since the parameters lie in Euclidean space.

**Dataset.** The CIFAR-10 dataset (Krizhevsky & Hinton, 2009) is a widely used benchmark in machine learning and computer vision. It consists of 60,000 color images, each of size $32 \times 32$ pixels, divided evenly into 10 distinct classes. Each class contains 6,000 images representing common objects including airplanes, cars, birds, cats, deer, dogs, frogs, horses, ships, and trucks. The dataset is split into 50,000 training images and 10,000 test images, enabling clear evaluation and benchmarking of image classification models. CIFAR-10 is known for its moderate size and diversity, making it suitable for rapid experimentation with machine learning algorithms such as convolutional neural networks (CNNs) and other classification methods. It was originally developed by researchers at the University of Toronto as a subset of the larger Tiny Images dataset, with manual labeling to ensure quality. Its balanced classes and diversity make it a challenging but accessible dataset for developing and testing image recognition models.

**Training.** We adopt the training protocol from (Tran et al., 2025c), using the Adam optimizer (Kingma & Ba, 2014) with learning rate set to 0.001. The total number of projections is fixed at 2,000 for all SW-based regularizers, and for TSW-based methods, we set the number of trees to 200.

**Architecture for MCTSW-VAE.**

**Encoder:**

$$x \in \mathbb{R}^{3 \times 32 \times 32} \rightarrow \text{Conv2d}_{32} \rightarrow \text{ReLU}$$
$$\rightarrow \text{Conv2d}_{32} \rightarrow \text{ReLU}$$
$$\rightarrow \text{Conv2d}_{64} \rightarrow \text{ReLU}$$
$$\rightarrow \text{Conv2d}_{64} \rightarrow \text{ReLU}$$
$$\rightarrow \text{Conv2d}_{128} \rightarrow \text{ReLU}$$
$$\rightarrow \text{Conv2d}_{128} \rightarrow \text{Flatten}$$
$$\rightarrow \text{FC}_{512} \rightarrow \text{ReLU} \rightarrow \text{FC}_9$$
$$\rightarrow \text{exp\_map} \rightarrow z \in \mathcal{M}$$

**Decoder:**

$$z \in \mathcal{M} \rightarrow \text{log\_map} \rightarrow \text{FC}_{512} \rightarrow \text{FC}_{2048} \rightarrow \text{ReLU}$$

### E.3 Graph Self-Supervised Learning

**Dataset.** Cora (Kipf & Welling, 2017) is a benchmark citation network where nodes are research papers and edges represent citation links. Each paper is represented by a binary bag-of-words vector (1 if a dictionary term appears, 0 otherwise), with labels corresponding to subject areas. We utilize the preprocessed version in DGL (Wang et al., 2020).

**Hyperparameter settings.** We use the Adam optimizer for all methods with learning rate $5 \times 10^{-4}$. The balancing hyperparameter $\lambda$ is selected from $\{0.01, 0.05, 0.1, 0.5, 1\}$ for best performance. The number of estimators for random forest is set to 200. Number of trees in each TSW-based method is 500 with 8 rays on a tree.

**Architecture.** We follow the same architecture as in (Liu et al., 2024) to use a GCN consist of 2 graph convolutional layer with hidden dimension of 256.

### E.4 Hardware Settings

All experiments were conducted on a single NVIDIA A100 (80GB) GPU server.

### E.5 Runtime Analysis

The training times for our three experiments involving variants of Wasserstein distance are summarized in Table 4, 5, and 6. It is worth noting that, in the gradient-flow setting (Table 4), our imple-

Table 4: Training time and performance for the gradient flow experiment.

| Method | SW_ambient | Prod-TSW | Prod-SW | MCSTSW |
|---|---|---|---|---|
| Time (s) | 158 | 617 | 468 | 303 |
| $\log W_2$ ($\downarrow$) | 0.33 | 0.34 | 0.31 | $\mathbf{-3.65}$ |

Table 5: Training time and accuracy for the graph self-supervised learning task.

| Method | S-TSW-SSGE | H-TSW-SSGE | E-TSW-SSGE | MC-TSW-SSGE |
|---|---|---|---|---|
| Time (s) | $63.15 \pm 0.60$ | $63.36 \pm 1.08$ | $63.06 \pm 0.95$ | $62.93 \pm 0.84$ |
| Accuracy ($\uparrow$) | $78.3 \pm 0.2$ | $75.1 \pm 0.2$ | $77.9 \pm 0.3$ | $\mathbf{79.9 \pm 0.5}$ |

mentations of Prod-TSW and Prod-SW apply TSW/SW independently to each subspace, whereas MCTSW exploits a highly parallelizable tree-based construction. This yields a substantially lower runtime for MCTSW despite operating in a mixed-curvature space. The same parallelization strategy explains why the MCTSW-based variants in Tables 5 and Table 6 achieve training times that are competitive with other latent-space baselines while achive superior performance.

### E.6 ABLATION STUDY

We provide ablation studies over these hyperparameters on graph self-supervised learning task in Tables 1 and 2. In our notation, $max_K = K_0 > 0$ and $m = m_0$ mean that we use $m_0$ subspaces whose curvatures are in $[-K_0, -K_0 + \frac{2K_0}{m_0}, -K_0 + \frac{4K_0}{m_0}, \ldots, K_0 - \frac{2K_0}{m_0}]$, which we implement in code as `np.arange(-m0/2, m0/2) * 2 * K0 / m0`. Note that $max_K = 0$ recovers the Euclidean space (E-TSW-SSGE in Table 2). The results in Table 7 and Table 8 suggest that moderate values of both the curvature magnitude and the number of subspaces are most beneficial.

For the parameter $k$ (number of rays per tree), we additionally conduct an ablation study on the graph self-supervised learning task with $max_K = 4$ and $m = 32$. The results in Table 9 show that the performance is stable across a wide range of $k$.

## F DESIGN CHOICES

### F.1 PROJECTION FUNCTION AND SPLITTING MAP

We implement three additional variants, covering different combinations of

- projection rules (including spherical projection (our version) and orthogonal projection, and
- mass-distribution strategies across rays (by distance to the orthogonal projection (our version) or to the spherical projection),

along with a detailed comparison of the time complexity for each method. Across all variants, the two fundamental tree factors—root position and ray directions—are incorporated carefully to ensure that no information is lost in projection. The performance and running time of these variants on the Gradient Flow task with the same setting as in Section 5.1 are shown in Table 10. It is evident that our version of MCTSW outperforms the three others and has lower run time.

Looking at the runtime, it is noteworthy that orthogonal projection requires a different sort for each line, resulting in a time complexity of $\mathcal{O}(Lkn \log n)$ instead of $\mathcal{O}(Ln \log n)$, as with the spherical projection in our design choice. Furthermore, the spherical splitting function, while having the same time complexity $\mathcal{O}(Ldmnk)$, has a higher hidden constant, leading to a significantly higher run time. Note that in our notation, $L$ is the number of trees used for Monte Carlo estimation, $k$ denotes the number of rays per tree, $n$ is the number of support points, $d$ denotes the dimension of each component, and $m$ is the number of components. In our experiments, we use $L = 360$, $d = 2$, $m = 3$, $n = 2400$, and $k = 5$.

Table 6: Training time per epoch and test BCE for the VAE task.

| Method | SWAE | STSW-VAE | HSW-VAE | MCSTSW-VAE |
|---|---|---|---|---|
| Time (s/epoch) | $9.64 \pm 0.09$ | $7.25 \pm 0.01$ | $9.76 \pm 0.18$ | $10.62 \pm 0.05$ |
| Test BCE ($\downarrow$) | $0.6043 \pm 0.0005$ | $0.6026 \pm 0.0009$ | $0.6012 \pm 0.0006$ | $\mathbf{0.6000 \pm 0.0002}$ |

Table 7: Ablation study on the maximum magnitude of curvature $K$ when $m = 32$.

| $max_K$ | 0 | 1 | 2 | 4 | 8 |
|---|---|---|---|---|---|
| Accuracy ($\uparrow$) | $77.9 \pm 0.3$ | $77.7 \pm 0.1$ | $78.3 \pm 0.2$ | $79.9 \pm 0.5$ | $77.3 \pm 0.5$ |

Moreover, we would like to note that the metricity properties can be verified similarly to the proof in Appendix B and Appendix C for the two versions using spherical projection, while the metricity of the two variants using orthogonal projection requires further investigation.

### F.2 TREE SIMPLIFICATION

We developed a version of MCTSW that sample trees without our simplification specified in Section 4 and conducted an additional experiment, comparing the two version. The only difference between the two lies in how the splitting function is computed. In our version, there exists a closed-form expression for the distance from a point to a geodesic ray (detailed in Appendix D.1), whereas the non-simplified version has to approximate this distance numerically.

For this case, we adopt ternary search algorithm (CP Algorithm) to approximate the $d_{\mathcal{M}}(z, r_x^{y_i}) = \min_{h \in \bar{r}_x^{y_i}} d_{\mathcal{M}}(h, z) = \min_{t \in \mathbb{R}^+} d_{\mathcal{M}}(z, x \oplus t \otimes ((-x) \oplus y),)$ and we terminate the search after 10 iterations.

The results on the gradient-flow experiment are reported in Table 11. The two versions achieve similar performance in terms of $\log W_2$, while the non-simplified variant is significantly slower in wall-clock time. This empirically supports our choice of the simplified formulation.

### F.3 TREE TOPOLOGY

we have further evaluated the impact of the tree topology by implementing an MCTSW variant in the non-simplified setting with a sequence (chain-like) topology, i.e., a sequence of lines joined at the intersection of every two consecutive lines. The trees are sampled by drawing the intersection points from a wrapped normal distribution centered at the origin.

As this variant consumes substantially more memory, we use a smaller gradient-flow configuration for a fair comparison between the two topologies, namely $L = 50$, $d = 2$, $m = 3$, $n = 100$, and $k = 5$. The results in Table 12 show that the star topology used in our main experiments clearly outperforms the chain-like topology in terms of $\log W_2$.

Table 8: Ablation study on the number of subspaces $m$ when $max_K = 4$.

| $m$ | 8 | 32 | 128 |
|---|---|---|---|
| Accuracy ($\uparrow$) | $76.7 \pm 1.1$ | $79.9 \pm 0.5$ | $78.7 \pm 0.2$ |

Table 9: Ablation study on the number of lines per tree $k$ when $max_K = 4$ and $m = 32$.

| $k$ | 4 | 8 | 16 | 32 | 64 | 128 | 256 | 512 |
|---|---|---|---|---|---|---|---|---|
| Accuracy ($\uparrow$) | $79.6 \pm 0.2$ | $79.9 \pm 0.5$ | $79.5 \pm 0.3$ | $79.6 \pm 0.4$ | $79.8 \pm 0.1$ | $79.7 \pm 0.4$ | $79.8 \pm 0.6$ | $79.5 \pm 0.3$ |

Table 10: Results on gradient-flow task $\log W_2$ and run time.

| Combination | $\log W_2$ ($\downarrow$) | Run time (s) |
|---|---|---|
| Orthogonal split + Orthogonal projection | $-1.7517$ | 383 |
| Orthogonal split + Spherical projection (ours) | $\mathbf{-3.6576}$ | **303** |
| Spherical split + Orthogonal projection | $-2.6038$ | 1350 |
| Spherical split + Spherical projection | $-3.5546$ | 1110 |

Table 11: Comparison of simplified and non-simplified MCTSW (gradient flow task).

| Method | $\log W_2$ ($\downarrow$) | Run time (s) |
|---|---|---|
| Simplified | $-3.6576$ | 303 |
| Non-simplified | $-3.6569$ | 3479 |

Table 12: Comparison of MCTSW with star topology and chain-like topology.

| Tree topology | $\log W_2$ ($\downarrow$) |
|---|---|
| Star | $-3.3788$ |
| Chain-like | $-1.7140$ |

