# OpenReview forum: "Mixed-Curvature Tree-Sliced Wasserstein Distance"
_ICLR.cc/2026/Conference — ICLR 2026 Poster_

### Official Review · Reviewer_So7Z · 2025-10-31

**Soundness:** 3
**Presentation:** 2
**Contribution:** 3
**Rating:** 4
**Confidence:** 2

**Summary:**

This paper proposes an algorithm to construct a tree when the data lies in a mixed-curvature space. Then, using this tree metric, this paper proposed computing the tree-wasserstein distance efficiently.

**Strengths:**

* This paper proposed a novel method to construct the tree.
* The proposed methods can capture the structure of Muxture-curvature spaces, and the effectiveness of the proposed methods was demonstrated in several tasks: gradient flow and self-supervised learning.

**Weaknesses:**

* This paper did not evaluate the approximation error of the proposed method and the original Wasserstein distance. For instance, can the proposed method more accurately approximate the Wasserstein distance than the tree-sliced Wasserstein distance with the tree metric constructed by the clustering method [4]? This paper evaluated the performance of the proposed method in several tasks; however, these results are influenced by various factors. It would be good if this paper could demonstrate the effectiveness of the proposed method in the simplest task.
* Various methods have been proposed for constructing tree metrics for the tree-Wasserstein distance, e.g., [1,2,3,4], but this paper did not discuss the relationship between the proposed method and these existing approaches. Although the proposed method appears to differ from these existing approaches, it is necessary to discuss its relationship to them.
* In the experiments, what methods did this paper use for TSW to construct the tree metric?
* This paper did not evaluate the computational efficiency of the proposed method. In lines 378-388, this paper discusses the time complexity; however, can this paper also report the time required for training in the experiments?

## Reference
[1] Indyk et. al., Fast image retrieval via embeddings. In International Workshop on Statistical and Computational Theories of Vision 2003

[2] Takezawa et. al., Supervised tree-wasserstein distance. In International Conference on Machine Learning 2021.

[3] Lin et. al., Tree-wasserstein distance for high dimensional data with a latent feature hierarchy, In International Conference on Learning Representation 2021.

[4] Le et. al., Tree-sliced variants of wasserstein distances, In Neural Information Processing Systems 2019

**Questions:**

See the weakness section.

---

> ### Author Response · Authors · 2025-11-23
> **Reply [1/2]**
>
> We thank the Reviewer for the response and address the concerns below. The Reviewer may also refer to our General Response for a summary of the revisions.
>
>
> ---
>
> **W1. This paper did not evaluate the approximation error of the proposed method and the original Wasserstein distance. For instance, can the proposed method more accurately approximate the Wasserstein distance than the tree-sliced Wasserstein distance with the tree metric constructed by the clustering method [4]? This paper evaluated the performance of the proposed method in several tasks; however, these results are influenced by various factors. It would be good if this paper could demonstrate the effectiveness of the proposed method in the simplest task.**
>
> **Answer W1.** We acknowledge that, for several variants of the Sliced Wasserstein distance, prior works have established approximation guarantees and upper/lower bounds relative to the original Wasserstein distance. However, to the best of our knowledge, no such bounds currently exist for any tree-sliced variant. The main challenge lies in the structural complexity of trees: unlike the one-dimensional projections used in classical sliced Wasserstein, tree structures are much harder to analyze, making it difficult to relate tree-induced projections to the geometry of the underlying Wasserstein space.
>
> For the clustering method proposed in [4], we kindly refer the Reviewer to **W2** below, where we clarify in detail why the approach in [4], as well as several mentioned methods, is not applicable in our setting.
>
>
> **W2. Various methods have been proposed for constructing tree metrics for the tree-Wasserstein distance, e.g., [1,2,3,4], but this paper did not discuss the relationship between the proposed method and these existing approaches. Although the proposed method appears to differ from these existing approaches, it is necessary to discuss its relationship to them.**
>
> **Answer W2.** We thank the Reviewer for raising this point, and we agree that our proposed method may appear similar to the approaches in [1,2,3,4] at first glance. We therefore clarify the relationship between these works and our tree-sliced approach. Although both lines of work share similar terminology and both exploit the closed-form expression of OT on tree metric spaces, they differ in a fundamental way. The distinction lies in **what is being sampled** and **whether the underlying measures have static or dynamic support**.
>
> **Tree-sliced OT in [1,2,3,4]: fixed tree space, static-support measures**
>
> - These works typically consider static-support measures arising in applications such as classification or topological data analysis.
>
> - The tree metric space is fixed: it is usually obtained once via a clustering-based construction (e.g., hierarchical clustering).
>
> - The “slicing” step involves sampling tree metrics on this fixed space, and approximating OT by averaging over these sampled metrics.
>
> - Crucially, because the data distribution does not change, the tree structure is constructed once and reused throughout.
>
> **Tree-sliced OT in [5,6]: sampled tree systems, dynamic-support measures**
>
> - These works are designed for dynamic-support measures, which commonly arise in generative modeling and training of implicit models. The support of the model distribution changes throughout optimization.
>
> - Here, the "tree system" itself is sampled (a set of rays forming a tree-like structure). Once sampled, the tree metric on that system is fixed.
>
> - The Wasserstein distance is computed on each sampled tree system, and the final tree-sliced distance is the average over sampled tree systems.
>
> **This distinction is essential:**
>
> - [1,2,3,4]: fix the space, sample the metric.
>
> - [5,6]: sample the space, and use the induced metric.
>
> Our work follows the setting of [5,6], where the support of the data distribution is changing continuously over training (e.g., when pushing a distribution toward an empirical target, its support evolves every iteration). In this regime, the clustering-based approach in [1,2,3,4] is not applicable, because:
>
> 1. One would need to recompute the clustering at every optimization step to adapt to the changing support.
>
> 2. This is computationally prohibitive.
>
> 3. Repeated re-clustering also does not yield a meaningful or stable geometric structure for guiding gradient-based optimization.
>
> Moreover, the tree-sliced approaches in [5,6] and in our work admit a closed-form approximation (see Equation (21)), which makes the method computationally efficient even in dynamic-support scenarios. For these reasons, both our approach and prior works operating in the dynamic-support regime [5,6] do not adopt the methods in [1,2,3,4]. The two lines of work address fundamentally different problem settings.

---

> ### Author Response · Authors · 2025-11-23
> **Reply [2/2]**
>
> **W3. In the experiments, what methods did this paper use for TSW to construct the tree metric?**
>
> **Answer W3.** The mixed-curvature tree, together with its topology and metric structure, is defined in Section 3.1. This space is constructed by gluing $k$ geodesic rays (each defined in Definition 3.1) at their common origin. The gluing procedure is described in lines 233–239. Intuitively, one may visualize this space as a star-shaped structure in which $k$ rays emanate from the same starting point in Euclidean space.
>
> The metric of this space is defined in Definition 3.3. Each ray is treated as a one-dimensional geodesic, and the distance between two points is simply the one-dimensional distance along the corresponding ray. The construction admits a natural geometric interpretation, explained in Remark 3.5; for convenience, we reproduce that explanation below.
>
> > The metric $d_\mathcal{T}$ encodes the tree structure within mixedcurvature geometry. Specifically, when two points lie on the same geodesic ray ($i = j$), their distance coincides with the ambient metric along that ray. Conversely, when two points lie on different rays ($i \neq j$), their distance is computed by traveling through the common root $x$, which links rays that may correspond to regions of differing curvature.
>
> **W4. This paper did not evaluate the computational efficiency of the proposed method. In lines 378-388, this paper discusses the time complexity; however, can this paper also report the time required for training in the experiments?**
>
> **Answer W4.** Thank you for your feedback regarding the evaluation of computational efficiency. In response, the training times for our three experiments involving variants of Wasserstein distance are summarized below. All experiments were conducted on a single A100 GPU.
>
> **Table 1: Training Time for Gradient Flow Experiment**
>
> | Method    | SW_ambient | Prod-TSW | Prod-SW | MCSTSW |
> |-----------|------------|----------|---------|--------|
> | Time (s)  | 158        | 617      | 468     | 303    |
> | $\log W_2 (\downarrow)$ | 0.33 | 0.34 | 0.31| −3.65  |
>
> **Table 2: Training Time for Graph Self-Supervised Learning Task**
>
> | Method       | S-TSW-SSGE   | H-TSW-SSGE   | E-TSW-SSGE   | MC-TSW-SSGE  |
> |--------------|--------------|--------------|--------------|--------------|
> | Time (s)     | $63.2 \pm 0.6$ | $63.4 \pm 1.1$ | $63.1 \pm 1.0$ | $63.0 \pm 0.8$ |
> | Accuracy $(\uparrow)$ | $78.3 \pm 0.2$ | $75.1 \pm 0.2$ | $77.9 \pm 0.3$ | $79.9 \pm 0.5$|
>
> **Table 3: Training Time for VAE Task**
>
> | Method       | SWAE         | STSW-VAE    | HSW-VAE     | MCSTSW-VAE     |
> |--------------|--------------|-------------|-------------|----------------|
> | Time (s/epoch)| $9.6 \pm 0.1$  | $7.3 \pm 0.1$ | $9.8 \pm 0.2$ | $10.6 \pm 0.1$   |
> | Test BCE ($\downarrow$)| $0.6043 \pm 0.0005$ | $0.6026 \pm 0.0009$ | $0.6012 \pm 0.0006$ | $0.6000 \pm 0.0002$
>
>
> It is worth noting that, in the gradient-flow setting (Table 1), our implementations of Prod-TSW and Prod-SW apply TSW/SW independently to each subspace, whereas MCTSW exploits a highly parallelizable tree-based construction. This yields a substantially lower runtime for MCTSW despite operating in a mixed-curvature space. The same parallelization strategy explains why the MCTSW-based variants in Tables 2 and Table 3 achieve training times that are competitive with other latent-space baselines while achive superior performance.
>
>
>
>
> *Reference*
>
> [1] Indyk et al., Fast image retrieval via embeddings. In International Workshop on Statistical and Computational Theories of Vision 2003
>
> [2] Takezawa et al., Supervised tree-wasserstein distance. In International Conference on Machine Learning 2021.
>
> [3] Lin et al., Tree-wasserstein distance for high dimensional data with a latent feature hierarchy, In International Conference on Learning Representation 2021.
>
> [4] Le et al., Tree-sliced variants of wasserstein distances, In Neural Information Processing Systems 2019
>
> [5] Tran et al., Distance-Based Tree-Sliced Wasserstein Distance
>
> [6] Tran et al., Tree-Sliced Wasserstein Distance: A Geometric Perspective
>
> ---
> We thank the Reviewer for the constructive feedback and thoughtful suggestions. If our responses adequately address the concerns, we kindly hope that the evaluation may be adjusted to reflect this. We remain open to further discussion during the next stage of discussion.

---

### Official Review · Reviewer_NuB5 · 2025-10-31

**Soundness:** 3
**Presentation:** 3
**Contribution:** 3
**Rating:** 8
**Confidence:** 3

**Summary:**

By utilizing mixed-curvature spaces that enable better data representations aligned with complex underlying structures—rather than relying solely on the Euclidean setting—this paper introduces the Mixed-Curvature Tree-Sliced Wasserstein, a computationally efficient discrepancy measure between distributions. The proposed technique builds upon the ideas behind the classical Sliced-Wasserstein (SW) framework, leading the authors to introduce a Radon-type transform. The paper demonstrates the effectiveness of this new tool through experiments involving generative flow models and variational autoencoders.

**Strengths:**

- The paper is very well organized.
- The paper is thorough and complete: it presents solid theoretical results, clearly introduces the necessary notions and new definitions, establishes key properties, and provides simplifications that enhance its applicability. It also includes extensive references to related work, provides time complexity analyses, and validates the proposed framework through different experiments.
- The idea is novel and has the potential to be applied to data analysis across various disciplines.

**Weaknesses:**

The only potential weakness is that the paper might appear overly technical at times; for example, it would benefit from including a brief explanation of how to obtain the closed-form expression in Equation (21).

**Questions:**

- Although I understand the general idea, I got lost and could not clearly identify where the stereographic projection $\rho_k$ is utilized in the methodology. Measures are assumed already in the hyperspher $F$ Poncare ball P
- Could the authors clarify what they mean by “consistency of operators across geometries” in line 155?
- Is there any intuition or guideline on how to choose the parameters $k$ and $m$?

Style Suggestions:
- The paper alternates between using MC-TSW and MCTSW. I suggest unifying the notation throughout the text.
- I would reserve the symbol $\rho$ exclusively for the projection in Equation (3). For instance, I recommend using a different symbol for the Sliced-Wasserstein operator in Equation (8), as well as later in Equation (18) and similar instances.

**Details Of Ethics Concerns:**

No concerns.

---

> ### Author Response · Authors · 2025-11-23
> **Reply [1/2]**
>
> We thank the Reviewer for the response and address the concerns below. The Reviewer may also refer to our General Response for a summary of the revisions.
>
>
> ---
>
> **W1. The only potential weakness is that the paper might appear overly technical at times; for example, it would benefit from including a brief explanation of how to obtain the closed-form expression in Equation (21).**
>
> **Answer W1.** Equation (21),  where we present its derivation in Appendix D.2, comes dỉrectly from the closed-form expression of OT problems on tree metric space.
>
> We provide the details of this closed-form expression of OT problems on tree metric space as follows.
>
> Let $\mathcal{T}$ be a rooted tree (as a graph) with non-negative edge lengths, and the ground metric $d_{\mathcal{T}}$, i.e., the length of the unique path between two nodes. Given two probability distributions $\mu$ and $\nu$ supported on nodes of $\mathcal{T}$, the Wasserstein distance with ground metric $d_{\mathcal{T}}$ has closed form as follows:
>
> $W_{d_{\mathcal{T}},1}(\mu,\nu)
> = \sum_{e \in \mathcal{T}} w_e \cdot | \mu(\Gamma(v_e)) - \nu(\Gamma(v_e)) |,$
>
> where $v_e$ is the endpoint of edge $e$ that is farther away from the tree root, $\Gamma(v_e)$ is the subtree of $\mathcal{T}$ rooted at $v_e$, and $w_e$ is the length of $e$.
>
> **Q1. Although I understand the general idea, I got lost and could not clearly identify where the stereographic projection $\rho_k$ is utilized in the methodology. Measures are assumed already in the hypersphere $F$ Poincare ball P.**
>
> **Q2. Could the authors clarify what they mean by “consistency of operators across geometries” in line 155?**
>
> **Answer Q1+Q2.** The stereographic projection is used to construct a unified coordinate representation that embeds Euclidean, hyperspherical, and hyperbolic spaces into a common framework. Working in this unified space allows us to define a single family of operators that apply consistently across all three geometries; this is what we intended by "consistency of operators across geometries". This construction is not only conceptually convenient but also practically important as it enables a high parallelized implementation of MCTSW across the components regardless of the underlying geometry. We have made this clearer in the revised version of the paper.

---

> ### Author Response · Authors · 2025-11-23
> **Reply [2/2]**
>
> **Q3. Is there any intuition or guideline on how to choose the parameters $k$ and $m$?**
>
> **Answer Q3.**
> *The choice of $K$ (curvature) and $m$ (number of subspaces)* is application- and data-dependent because the latent geometry should match the underlying structure of the data. Intuitively, a larger magnitude of curvature makes the space more curved and therefore more different from Euclidean space, which can better capture non-Euclidean structure but may hurt when the data are closer to Euclidean. Consequently, we treat $K$ and $m$ as hyperparameters to be tuned on a validation set.
>
> We provide ablation studies over these hyperparameters in Tables 1 and 2. In our notation, $max_K = K_0 > 0$ and $m = m_0$ mean that we use $m_0$ subspaces whose curvatures are in $[-K_0,\,-K_0 + \tfrac{2K_0}{m_0},\, -K_0 + \tfrac{4K_0}{m_0},\,\ldots,\,K_0 - \tfrac{2K_0}{m_0}]$, which we implement in code as `np.arange(-m0/2, m0/2) * 2 * K0 / m0`. Note that $max_K=0$ recovers the Euclidean space (E-TSW-SSGE in Table 2 of our maintext). The results in Tables 1 and 2 below suggest that moderate values of both the curvature magnitude and the number of subspaces are most beneficial.
>
>
> *Table 1: Ablation study on the curvature $(K)$ when $m=32$.*
>
> | $max_K$  | 0  | 1          | 2          | 4          |  8          |
> |--------|------------|------------|------------|------------|------------|
> | Accuracy $(\uparrow)$   | $77.9\pm0.3$| $$77.7 \pm 0.1$$ | $$78.3 \pm 0.2$$ | $$79.9 \pm 0.5$$ | $$77.3 \pm 0.5$$ |
>
> *Table 2: Ablation study on the number of subspaces $(m)$ when $max_K=4$.*
>
> | m      | 8          | 32         | 128        |
> |--------|------------|------------|------------|
> | Accuracy $(\uparrow)$ | $$76.7 \pm 1.1$$ | $$79.9 \pm 0.5$$ | $$78.7 \pm 0.2$$ |
>
> *For the parameter $k$ (number of rays per tree)*, we additionally conduct an ablation study on the graph self-supervised learning task with $max_K=4$ and $m=32$. The results in Table 3 show that the performance is stable across a wide range of $k$ values.
>
> *Table 3: Ablation study on the number of lines per tree $(k)$ when $max_K=4$ and $m=32$.*
> |k|4|8|16|32|64|128|256|512|
> |-|-|-|-|-|-|-|-|-|
> |Accuracy $(\uparrow)$| $79.6\pm0.2$ | $79.9\pm0.5$ | $79.5\pm0.3$ | $79.6\pm0.4$ | $79.8\pm0.1$ | $79.7\pm0.4$ | $79.8\pm0.6$ | $79.5\pm0.3$|
>
> **Answer for style suggestions.**
> We appreciate the reviewer's careful reading and helpful style comments. In the revised manuscript, we now consistently use the notation MCTSW throughout. In addition, we reserve the symbol $\rho$ exclusively for the projection map in Eq. (3) and have updated the notation in the Sliced-Wasserstein operator (e.g., Eqs.(8) and (18)) so that the underlying measures are denoted by $\mu$ and $\nu$, for standardization and improved readability.
>
> ---
> We thank the Reviewer for the constructive feedback and thoughtful suggestions. We remain open to further discussion during the next stage of discussion.

---

> > ### Comment · Reviewer_NuB5 · 2025-11-25
> >
> > I thank the authors for their clear answers. I maintain my scores.

---

### Official Review · Reviewer_HAxH · 2025-11-01

**Soundness:** 3
**Presentation:** 3
**Contribution:** 3
**Rating:** 6
**Confidence:** 4

**Summary:**

The central hypothesis is that the Tree-Sliced Wasserstein framework can be generalized to mixed curvature spaces (MCS). The core technical contributions include a mixed-curvature tree system $\mathcal{T}$ and a Radon Transform $\mathcal{R}^{\alpha}$ for MCS, which maps mass from the manifold $\mathcal{M}$ onto $\mathcal{T}$. The authors claim this construction yields a closed-form, computationally efficient solution for the 1-Wasserstein distance on the tree. They propose that this MC-TSW metric will more faithfully capture the joint geometric structure of MCS distributions than separable baselines (like product-space SW/TSW), leading to superior performance.

**Strengths:**

- The paper is nicely written and tackles a well-motivated and non-trivial problem. The authors correctly identifies the limitations of naive baselines. That is, ambient-space metrics ignore the geometry, while separable "product" metrics (like Prod-TSW) ignore joint structure across components. This work appears to be the first to build a TSW-like distance specifically for these joint spaces.


- Good experimental results. For example,

  - the gradient flow experiment provides strong evidence for the paper’s central claim. In Table 1, the MC-TSW method successfully converges to a low $\log W_2$ error ($-3.65$). In stark contrast, the separable baselines, Prod-SW and Prod-TSW, fail entirely (positive $\log W_2$ errors). This result seems to suggest that the target distribution have cross-component correlations that only a joint metric like MC-TSW can capture.


  - the VAE and graph SSL experiments (Tables 2 & 3) further show that combining the proposed MC-TSW regularizer with an MCS latent space yields SOTA or highly competitive results, outperforming both KL-based regularizers and single-curvature (Euclidean, spherical, hyperbolic) models. This validates both the new metric and the underlying geometric hypothesis.

- The algorithm has the same asymptotic cost as TSW, parallelizes well on GPUs.

**Weaknesses:**

- The Radon transform (def 3.6) projects every point $z \in \mathcal{M}$ to the same coordinate $t = d_{\mathcal{M}}(x, z)$ on all $k$ rays of the tree. The only thing distinguishing the projection of $z$ onto different rays is the splitting map $\alpha(z, \mathcal{T})_i$. This spherical projection collapses all angular and component-specific directional information of $z$ relative to the root $x$, reducing it to a single scalar distance. This appears to be a geometrically lossy projection, and its trade-off is not discussed.

- Moreover, the paper's construction is limited to star-shaped trees at a single point $x$. The implementation in algo 1 simplifies this further by sampling rays that are axis-aligned, i.e., each ray extends in exactly one component $j$ of the product space. While the projection onto this tree (via the joint distance $d_{\mathcal{M}}$ and splitting map $\alpha$) is still joint, the tree structure itself is highly constrained and separable. This axis-aligned, star-shaped probe may not be the most effective geometry for capturing complex, non-axis-aligned correlations, even if it is computationally convenient.

- The $O(L n \log n)$ component of the complexity is a direct result of this information-collapsing projection. Because all points $a_j$ are projected to the *same* coordinate $c_j = d_{\mathcal{M}}(x, a_j)$ on every ray, a single sorting of the $n$ values $\{c_j\}$ is sufficient to compute the $W_1$ on the tree (Eq. 21). This tractability is thus achieved by collapsing all per-ray projection information into a single scalar distance.

**Questions:**

- The success in Table 1 suggests the joint projection (via $d_{\mathcal{M}}$ and $\alpha$) is doing all the work.  Does this imply the tree structure itself is less important than the projection function?

---

> ### Author Response · Authors · 2025-11-23
> **Reply [1/3]**
>
> We thank the Reviewer for the response and address the concerns below. The Reviewer may also refer to our General Response for a summary of the revisions.
>
>
> ---
>
>
> **W1. The Radon transform (def 3.6) projects every point $z\in\mathcal{M}$ to the same coordinate $t=d_\mathcal{M}(x,z)$ on all $k$ rays of the tree. The only thing distinguishing the projection of $z$ onto different rays is the splitting map $\alpha(z,\mathcal{T})_i$. This spherical projection collapses all angular and component-specific directional information of $z$ relative to the root $x$, reducing it to a single scalar distance. This appears to be a geometrically lossy projection, and its trade-off is not discussed.**
>
> **W3. The $O(Ln \log n)$ component of the complexity is a direct result of this information-collapsing projection. Because all points $a_j$ are projected to the *same* coordinate $c_j=d_\mathcal{M}(x,a_j)$ on every ray, a single sorting of the $n$ values $c_j$ is sufficient to compute the $W_1$ on the tree (Eq. 21). This tractability is thus achieved by collapsing all per-ray projection information into a single scalar distance.**
>
>
> **Answer W1+W3 (1).** Indeed, the two key ingredients in the Radon Transform defined in Definition 3.6 (Eq. 17) are:
>
> - Each point $z \in \mathcal{M}$ is projected onto every ray of the tree at the same radial coordinate $t = d_\mathcal{M}(x,z)$. This depends only on the distance from $z$ to the root $x$, and not on the specific ray direction.
> - At that radial location, the mass at $z$ is distributed across rays according to its relative distances to the rays. This depends on the entire tree geometry, including both the root and all ray directions.
>
> We chose the first design - placing all projections at the same radial coordinate $t = d_\mathcal{M}(x,z)$ - primarily for computational efficiency. With this choice, the sorting step only needs to be performed on one ray (see lines 369–377). The Reviewer did note this in **W3**, and this efficiency consideration is precisely our motivation for using this projection rule, instead of the more natural geodesic projection (analogous to orthogonal projection in Euclidean space). The latter would require sorting on all $k$ rays, increasing the computational cost.
>
> This same projection choice is also used in prior work on spherical data, e.g. [1], where all projected copies of a point share the same radial coordinate across tree edges.
>
> Reference:
>
> [1] Tran et al., Spherical Tree-Sliced Wasserstein Distance, ICLR 2024

---

> ### Author Response · Authors · 2025-11-23
> **Reply [2/3]**
>
> **Answer W1+W3 (2).** *Reviewer suggestion.* We appreciate the Reviewer’s comment regarding alternative projection schemes. In response, we implements three additional variants, covering different combinations of:
>
> - projection rules (including spherical projection (our version), orthogonal projection, and
> - mass-distribution strategies across rays (by distance to the orthogonal projection (our version) or spherical projection),
>
> along with a detailed comparison of the time complexity for each method. Across all variants, the two fundamental tree factors -- root position and ray directions -- are incorporated carefully to ensure that no information is lost in projection. The performance and running time of these variants on the gradient flow task with the same setting as in Section 5.1 are shown in Table 1. It is evident that our version of MCTSW outperforms the three others and has a lower run time.
>
> *Table 1. Results on gradient flow task $log W_2$ and run time.*
>
> | Combination                                   | $\log W_2 (\downarrow)$            | Run Time (s)      |
> |-----------------------------------------------|-------------------------|-------------------|
> | Orthogonal split + Orthogonal projection     | -1.7517                 | 383               |
> | Orthogonal split +  Spherical projection (ours) | **-3.6576**     | **303**               |
> | Spherical split + Orthogonal projection      | -2.6038                 | 1350              |
> | Spherical split + Spherical projection | -3.5546                 | 1110              |
>
> Regarding the runtime, we emphasize that the variants using orthogonal projection require an independent sort for each ray, leading to a higher runtime and a higher time complexity of $\mathcal{O}(L k n \log n)$, compared to $\mathcal{O}(L n \log n)$ for the spherical projection used in our design. Furthermore, although the spherical splitting function has the same asymptotic complexity $\mathcal{O}(L d m n k)$ as the orthogonal splitting counterpart, it incurs a larger constant factor, which translates into a substantially higher wall-clock runtime. Recall that in our notation $L$ is the number of trees used for Monte Carlo estimation, $k$ is the number of rays per tree, $n$ is the number of support points, $d$ is the dimension of each component, and $m$ is the number of components. In our experiments, we use $L = 360$, $d = 2$, $m = 3$, $n = 2400$, and $k = 5$ and $10000$ training step.
>
> Moreover, the metricity properties for the two variants using spherical projection can be established by arguments analogous to those in Appendices B and C. In contrast, the metricity of the two variants based on orthogonal projection is not immediate from our current analysis and remains an interesting direction for further investigation.

---

> ### Author Response · Authors · 2025-11-23
> **Reply [3/3]**
>
> **W2. Moreover, the paper's construction is limited to star-shaped trees at a single point $x$. The implementation in algo 1 simplifies this further by sampling rays that are axis-aligned, i.e., each ray extends in exactly one component $j$ of the product space. While the projection onto this tree (via the joint distance $d_\mathcal{M}$ and splitting map $\alpha$) is still joint, the tree structure itself is highly constrained and separable. This axis-aligned, star-shaped probe may not be the most effective geometry for capturing complex, non-axis-aligned correlations, even if it is computationally convenient.**
>
> **Q1. The success in Table 1 suggests the joint projection (via $d_\mathcal{M}$ and $\alpha$) is doing all the work. Does this imply the tree structure itself is less important than the projection function?**
>
> **Answer W2+Q1.** The reviewer raised questions about two other design choices, namely the simplification in tree sampling and tree topology.
>
> *Regarding the tree simplification*, we have additionally developed a version of MCTSW that sample trees without our simplification and conducted an additional experiment, comparing the two version. The only difference between the two lies in how the splitting function is computed. In our version, there exists a closed-form expression for the distance from a point to a geodesic ray (detailed in Appendix D.1), whereas the non-simplified version has to approximate this distance numerically.
>
> For the general case, we adopt ternary search algorithm [2] to approximate the $d_\mathcal{M}(z, r_x^{y_i}) = \min_{h\in \overline{r}_x^{y_i}} d_\mathcal{M}(h, z) = \min_{t\in\mathbb{R}^+} d_\mathcal{M}(z, x \oplus t\otimes((-x)\oplus y),)$ and we terminate the search after 10 iterations.
>
> The results on the gradient-flow experiment are reported in Table 2. The two versions achieve similar performance in terms of $\log W_2$, while the non-simplified variant is slower in wall-clock time. This empirically supports our choice of the simplified formulation.
>
> *Table 2: Comparison of simplified and non-simplified MCTSW (gradient flow task).*
> || $\log W_2 (\downarrow)$            | Run Time (s)    |
> |-|-|-|
> |Simplified| -3.6576 | 303 |
> |Non-simplified|-3.6569| 3479 |
>
> *Regarding tree topology*, we have further evaluated the impact of the tree topology by implementing an MCTSW variant in the non-simplified setting with a sequence (chain-like) topology, i.e., a sequence of lines joined at the intersection of every two consecutive lines. The trees are sampled by drawing the intersection points from a wrapped normal distribution centered at the origin.
>
> As this variant consumes substantially more memory, we use a smaller gradient-flow configuration for a fair comparison between the two topologies, namely $L=50$, $d=2$, $m=3$, $n=100$, and $k=5$. The results in Table 3 show that the star topology used in our main experiments clearly outperforms the chain-like topology in terms of $\log W_2$.
>
> *Table 3: Comparison of MCTSW with star topology and chain-like topology  (gradient flow task).*
> |Tree topology| $\log W_2 (\downarrow)$ |
> |-|-|
> |Chain-like|-1.7140|
> |Star| -3.3788 |
>
> Reference:
>
> [2] CP Algorithm. Ternary search. https://cp-algorithms.com/num_methods/ternary_search.html.
>
> ---
> We thank the Reviewer for the constructive feedback and thoughtful suggestions. If our responses adequately address the concerns, we kindly hope that the evaluation may be adjusted to reflect this. We remain open to further discussion during the next stage of discussion.

---

### Author Response · Authors · 2025-11-23
**General Response**

Dear AC and reviewers,

Thank you for your thoughtful reviews and valuable comments, which have greatly helped us improve the paper. We are encouraged by the positive feedback on several aspects of our work:

1. The problem setting was viewed as well-motivated and non-trivial (Reviewer HAxH), and our approach was recognized as novel (Reviewers So7Z, NuB5),

2. The method was praised as theoretically solid, well organized and noted as having potential for applications across different domains (Reviewer NuB5), and

3. The empirical evaluation was seen as providing strong evidence for the effectiveness of the proposed metric (Reviewers HAxH, NuB5).

Incorporating the reviewers' comments and suggestions, as well as additional empirical studies that we believe are informative, we summarize below the main revisions made to the manuscript (all changes are highlighted in red).

1. We added new experiments and analysis to compare alternative designs and to justify our design choices (splitting map, projection rule, simplification, and tree topology) in Appendix F.

2. We added an ablation study on the parameters $K, m$, and $k$ in Appendix E.6.

3. We added a run time analysis in Appendix E.5.

4. We added further clarification for the stereographic projection (lines 148-150, 159-161), closed-form formula in Equation (21) (lines 341-349), and extended the discussion for related work (lines 212-215).

---

### Meta-Review · Area_Chair_215X · 2026-01-06

**Summary:**

The paper builds novel method for Tree-Sliced Wasserstein distances on Mixed-Curvature spaces. The problem is non-trivial and the authors succesfully demonstrated the effectiveness and soundness of their approach. Additions propsed by Authors on different design choices for the tree construction, ablation studies and runtime comparisons  seem to answer concerns raised by the reviewers, therefore I am recommending this work for acceptance in ICLR program.

**Reviewer Concerns:**

Most of the concerns have been addressed by the rebuttal

**Reviewer Scores:**

Reviewer So7Z might have raised his score given the answers.

---

### Decision · Program_Chairs · 2026-01-26

Accept (Poster)